# Nucleotide resolution mapping of influenza A virus nucleoprotein-RNA interactions reveals RNA features required for replication

Graham D. Williams[1,2], Dana Townsend[2], Kristine M. Wylie[3,4], Preston J. Kim[5], Gaya K. Amarasinghe[5,6], Sebla B. Kutluay[2] & Adrianus C.M. Boon[1,2,5]

Influenza A virus nucleoprotein (NP) association with viral RNA (vRNA) is essential for packaging, but the pattern of NP binding to vRNA is unclear. Here we applied photo-activatable ribonucleoside enhanced cross-linking and immunoprecipitation (PAR-CLIP) to assess the native-state of NP–vRNA interactions in infected human cells. NP binds short fragments of RNA (~12 nucleotides) non-uniformly and without apparent sequence specificity. Moreover, NP binding is reduced at specific locations within the viral genome, including regions previously identified as required for viral genome segment packaging. Synonymous mutations designed to alter the predicted RNA structures in these low-NP-binding regions impact genome packaging and result in virus attenuation, whereas control mutations or mutagenesis of NP-bound regions have no effect. Finally, we demonstrate that the sequence conservation of low-NP-binding regions is required in multiple genome segments for propagation of diverse mammalian and avian IAV in host cells.

[1] Department of Medicine at Washington University School of Medicine, St Louis, MO 63110, USA. [2] Department of Molecular Microbiology at Washington University School of Medicine, St Louis, MO 63110, USA. [3] Department of Pediatrics at Washington University School of Medicine, St Louis, MO 63110, USA. [4] The McDonnell Genome Institute at Washington University School of Medicine, St Louis, MO 63110, USA. [5] Department of Pathology and Immunology at Washington University School of Medicine, St Louis, MO 63110, USA. [6] Department of Biochemistry and Biophysics at Washington University School of Medicine, St Louis, MO 63110, USA. Correspondence and requests for materials should be addressed to A.C.M.B. (email: jboon@wustl.edu)

nfluenza A virus (IAV) possesses a segmented, negative-sense
RNA genome that is bound by the viral nucleoprotein (NP)
throughout replication. Recent cryo-electron microscopy stu-
dies of the IAV ribonucleoprotein complex (vRNP) provide evi-
dence for a NP–RNA complex structure with a corkscrew-like
morphology and the tripartite polymerase complex at one end
and a loop at the other end. However, current models conflict
with each other and yield no information about RNA con-
formation, binding, or NP–RNA association[1,2]. NP is thought to
coat viral RNA (vRNA) uniformly in cells and virus particles;
however. uniform coating likely would preclude the possibility for
RNA structure formation in RNPs. Numerous virus families
utilize structured RNA elements for specific biological processes
throughout infection, including genome packaging[3,4]. For
example, viral RNA elements are required for efficient replication,
mRNA splicing, and genome packaging of IAV[5–7]. Structure
formation has been demonstrated with in vitro folded IAV vRNA
and engineered genome segments, but the structural constraints
imposed by nucleoprotein on vRNA generated during infection is
not known[8,9]. Elucidation of the physiological interaction
between NP and viral genomic RNA may provide novel insights
into how IAV is capable of coordinating its lifecycle. Thus, we set
out to determine the in vivo landscape of NP–vRNA interactions.

Infection and complete replication of IAV requires delivery of
all eight genome segments into a recipient cell. All IAV segments
require packaging signals derived from the termini on each seg-
ment[10,11]. Interaction between vRNAs has been demonstrated
in vitro and disruption of packaging signals or interacting seg-
ment regions attenuated virus replication at the stage of genome
packaging[12–14]. In many cases, mutation of a single segment leads
to a significant decrease in the packaging efficiency of other
segments[5,15]. Additionally, viral particles largely package only one
copy of each genome segment[16–18]. Together, these results sug-
gest that genome segments function as a multipartite, coopera-
tively packaged entity, possibly potentiated by segment–segment
interactions, rather than a stochastically generated particle[19,20].

In this study, we set out to determine how IAV NP interacts
with vRNA during infection in cells. We show that the NP of IAV
binds the vRNA non-uniformly and that regions of low-NP
binding are enriched for predicted RNA secondary structures.
Synonymous mutations designed to destabilize the predicted
RNA structure attenuate IAV replication, whereas synonymous
mutations that maintain the predicted RNA structure or muta-
tions in NP-bound RNA regions have no effect on virus repli-
cation in vitro or in vivo. Viral attenuation is associated with an
increase in defective virus production, suggesting that low-NP-
binding regions and the predicted RNA structures are required
for viral genome packaging.

## Results

**Nucleotide resolution mapping of NP–vRNA interactions**.
Photoactivatable ribonucleoside enhanced cross-linking and
immunoprecipitation (PAR-CLIP) coupled to next-generation
sequencing was used to resolve the interaction between the
negative-sense RNA genome of IAV and NP during infection of
human 293T cells[21]. We infected human cells with WT-PR8 virus
for 16 h in the presence of 4-thiouridine (4-SU) to enhance cross-
linking of NP–RNA complexes and then generated Illumina
$1 \times 50$ sequencing libraries of the NP-bound RNA (Fig. 1a). The
impact of 4-SU on viral replication was assessed in 293T cells.
WT-PR8 replicated to equivalent titers 12, 18, and 24 h post-
infection (hpi) in mock- or 4-SU-treated (100 μM) cells (Fig. 1b).
Additionally, NP localization after 4-SU treatment was assessed
by confocal microscopy at 16 hpi, and no alteration was observed
at this time point (Fig. 1b). These results suggest that 4-SU

treatment does not substantially impact IAV nucleoprotein pro-
duction or replication in human cells.

To determine the sensitivity and specificity of the PAR-CLIP
assay, we performed Western blotting analysis for IAV NP and
cellular ß-actin on input lysate and immunoprecipitated proteins
(Fig. 1c, bottom). Compared to immunoprecipitations performed
without antibody or a control anti-HA antibody, immunopreci-
pitation with a monoclonal antibody (MAb) against IAV NP
produced a specific band. UV-exposure of infected cells in the
presence of 4-SU enabled greater recovery of NP–RNA complexes
(Fig. 1c, top). The protein purity in the immunoprecipitate was
verified by silver stain or Western blotting with an anti-IAV
polyclonal serum (Supplementary Fig. 1a, b).

**Influenza A nucleoprotein binds viral RNA non-uniformly**.
PAR-CLIP identified both human and virus-derived RNAs that
interacted with NP. The procedure enriched for IAV RNA
sequences relative to RNA-seq libraries (Fig. 1d), and the majority
of the viral RNA sequences were derived from the negative-strand
vRNA (Supplementary Fig. 2). The average length of the NP-
bound vRNA was 12 nucleotides (range 11–14, Fig. 1e).

We then compared PAR-CLIP and RNA-seq libraries to
identify contiguous regions of vRNA that are significantly under-
and overrepresented among the NP-bound RNAs. Using the
criteria of >3-fold difference, $Q < 0.01$, and ≥ 18 nucleotides long,
we identified 24 regions in the viral genome that were low in NP
binding relative to RNA-seq and 18 regions that met two of these
three criteria that we did not investigate further (Supplementary
Fig. 3, Supplementary Table 1). Moreover, four high-NP-binding
regions where vRNA was overrepresented in PAR-CLIP libraries
relative to the control RNA-seq sets were identified (Supplemen-
tary Table 1). The low-NP-binding regions together represent
~10% of the viral genome and do not differ in base composition
from the remainder of the genome (Fig. 1f). Analysis of the
nucleotide sequences in low-NP-binding regions revealed that
RNA secondary structures are predicted to form in the absence of
NP binding in some of these regions (Supplementary Table 1)[5,7].
Thus, NP binding might be affected by local secondary structures
in the genome of IAV.

**Segment 5 low-NP-binding regions are important for IAV**. To
assess the significance of the NP-bound or underrepresented
vRNA regions, we selected six regions in segment 5 that have
variable NP-binding profiles (Fig. 2a and Table 1). The lack of
alternative reading frames and splicing of segment 5 mRNA made
this segment highly suitable for extensive characterization of the
significance of the low-NP-binding regions on IAV replication.
Of the six regions, two were underrepresented (low-NP binding)
in PAR-CLIP data sets ($NP_{22–68}$ and $NP_{1410–1495}$). The
$NP_{1410–1495}$ region contains a previously hypothesized vRNA
pseudoknot[5]. Additional regions of NP intermediate binding
($NP_{145–175}$, $NP_{456–490}$, $NP_{584–608, \text{ and }}$ $NP_{1058–1081}$) were also
included in the following studies. Computational prediction of
RNA structures using RNAfold in low-NP-binding regions gui-
ded further mutational analysis, and these regions were muta-
genized to either disrupt ($NP_{22–68:A}$ and $NP_{1410–1495}$) or maintain
($NP_{22–68:B}$) the predicted secondary and tertiary structure for-
mation (Supplementary Fig. 4). Mutant viruses bearing
2–7 synonymous structural nucleotide changes in these regions
were generated and assessed for ability to replicate in vitro and
in vivo.

WT and mutant viruses were subjected to a focus-forming
assay in MDCK cells to determine replication competence in vitro
(Fig. 2b). Mutations disrupting the segment 5 vRNA pseudoknot
($NP_{1410-1495}$) formation and destabilizing a predicted stem-loop

structure ($NP_{22–68:A}$) in the 3′ region of the vRNA segment resulted in reduction of focus area, as a measure of multi-cycle replication and spread (Fig. 2b). Conversely, mutation of intermediate NP-bound vRNA regions did not alter the focus area. Synonymous mutations in $NP_{22–68}$ designed to maintain the predicted secondary structure ($NP_{22–68:B}$), also did not affect focus size (Fig. 2b). Multi-cycle replication assays in MDCK cells of select mutant viruses confirmed these results (Fig. 2c). Finally, mice were inoculated with $10^3$ $TCID_{50}$ of each virus, and the presence of infectious virus in the lung was assessed 48 h later. WT-PR8 replicated to high titers at this time point whereas destabilizing mutations in low-NP-binding regions resulted in attenuation (Fig. 2d). Synonymous structural mutations in NP-bound regions had no effect on virus infection in vivo (Fig. 2d).

These results collectively suggest that structural features of the low-NP-binding regions are important for IAV replication.

To determine the cause of attenuated replication of low-NP-binding mutant viruses, we evaluated effects on specific stages in the IAV lifecycle. All viruses displayed similar cytoplasmic distribution of NP 8 hpi in MDCK cells when assessed by indirect immunofluorescence and confocal microscopy (Fig. 3a). The ability of the NP proteins from mutant viruses to facilitate the transcription and replication of a firefly luciferase reporter genome segment by the tripartite IAV polymerase complex was tested in human cells; all viruses displayed equivalent reporter activity (Fig. 3b). Similarly, infection of MDCK cells with all viruses generated equivalent amounts of viral antigen (NP) 8 hpi when measured by flow cytometry (Fig. 3c). Therefore, the

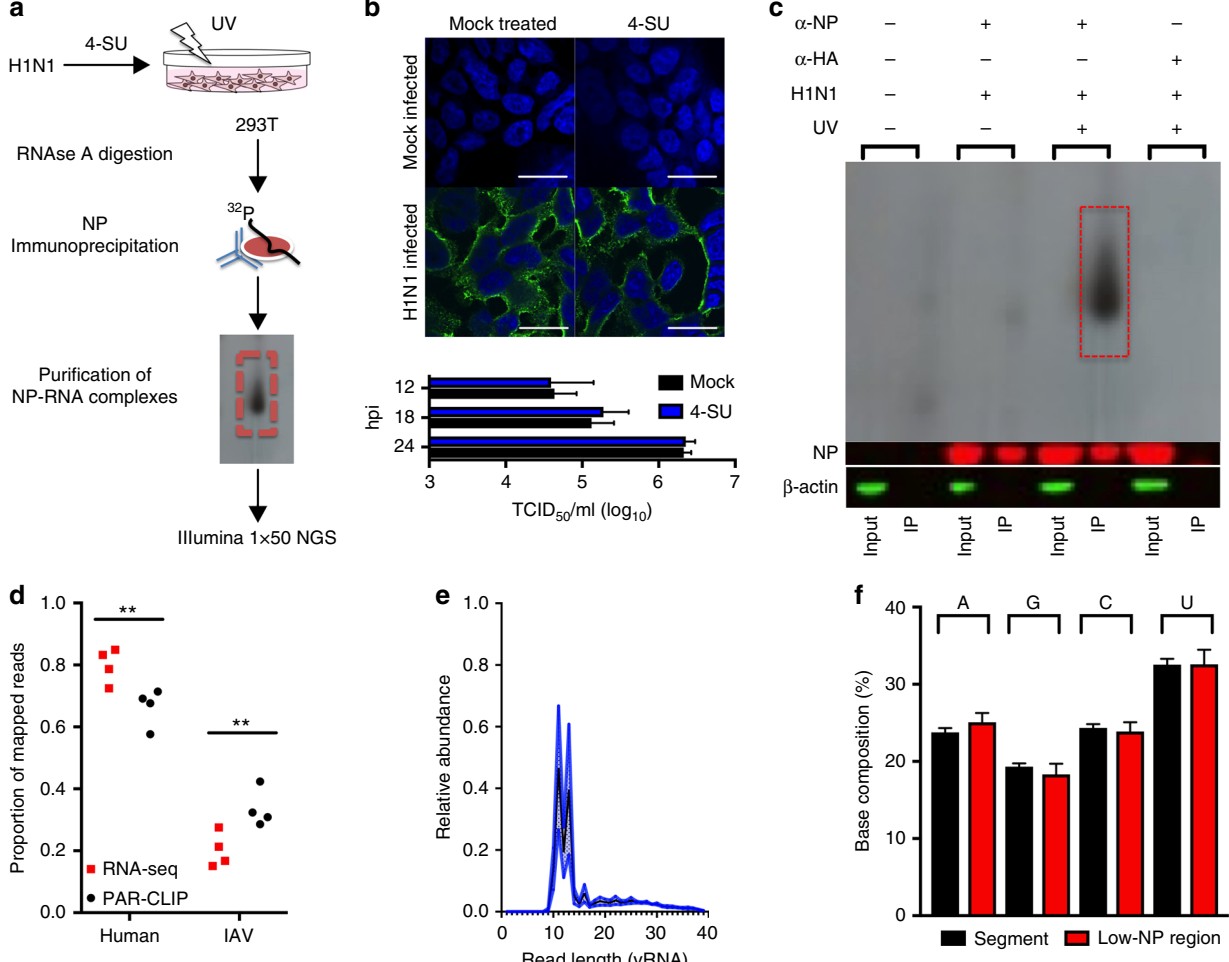

**Fig. 1** Development of PAR-CLIP for IAV NP. **a** Schematic for IAV NP PAR-CLIP assay. **b** Effects of 4-SU on IAV replication. Viral replication (MOI = 0.1) in the presence or absence of 4-SU (100 μM) was assessed by performing a growth curve at the indicated times in 293T cells and titered by $TCID_{50}$ assay in MDCK cells (bottom). Results are the average + s.e.m. of two experiments. NP localization was assessed following treatment and infection of 293T cells by confocal microscopy (top). Immunofluorescence staining for NP (green) was assessed in the presence or absence of 4-SU and counterstained with DAPI to identify cellular nuclei (blue). Scale bar indicates 25 μm. **c** PAR-CLIP was conducted on 293T cells infected with WT-PR8 in the absence (lanes 1 and 2) or presence of a monoclonal antibody against to IAV NP (lanes 3–6) or viral hemagglutinin protein (HA) (lanes 7–8). The effect of UV cross-linking on binding of RNA to the viral NP is shown in lanes 3–4 (no UV) and lanes 5–6 (with UV). Radioactivity ($^{32}P$) is visualized by autoradiograph and the presence or absence of NP and cellular ß-actin was done by western blot. The input sample and eluate are loaded in the uneven and even lanes, respectively. The results are representative of four independent experiments. Original western blots and autoradiograph are shown in Supplementary Fig. 7. **d** Proportion of PAR-CLIP or RNA-seq derived reads mapping to human or IAV genomes (**$P < 0.01$ by one-way ANOVA with multiple comparisons correction (Kruskal–Wallis test), $n = 4$). **e** Length of negative-sense viral RNA (vRNA) aligning reads was determined using FastX Toolkit and the number of reads of a certain length is plotted as a proportion of total vRNA mapping reads. Mean (black line) ± s.e.m. (blue shading) of 4 experiments. **f** Nucleotide composition of low-NP-binding regions and IAV genome (displayed as average + s.e.m. base composition of all eight gene-segments). No significant differences were detected between groups (one-way ANOVA with multiple comparisons correction (Kruskal–Wallis test))

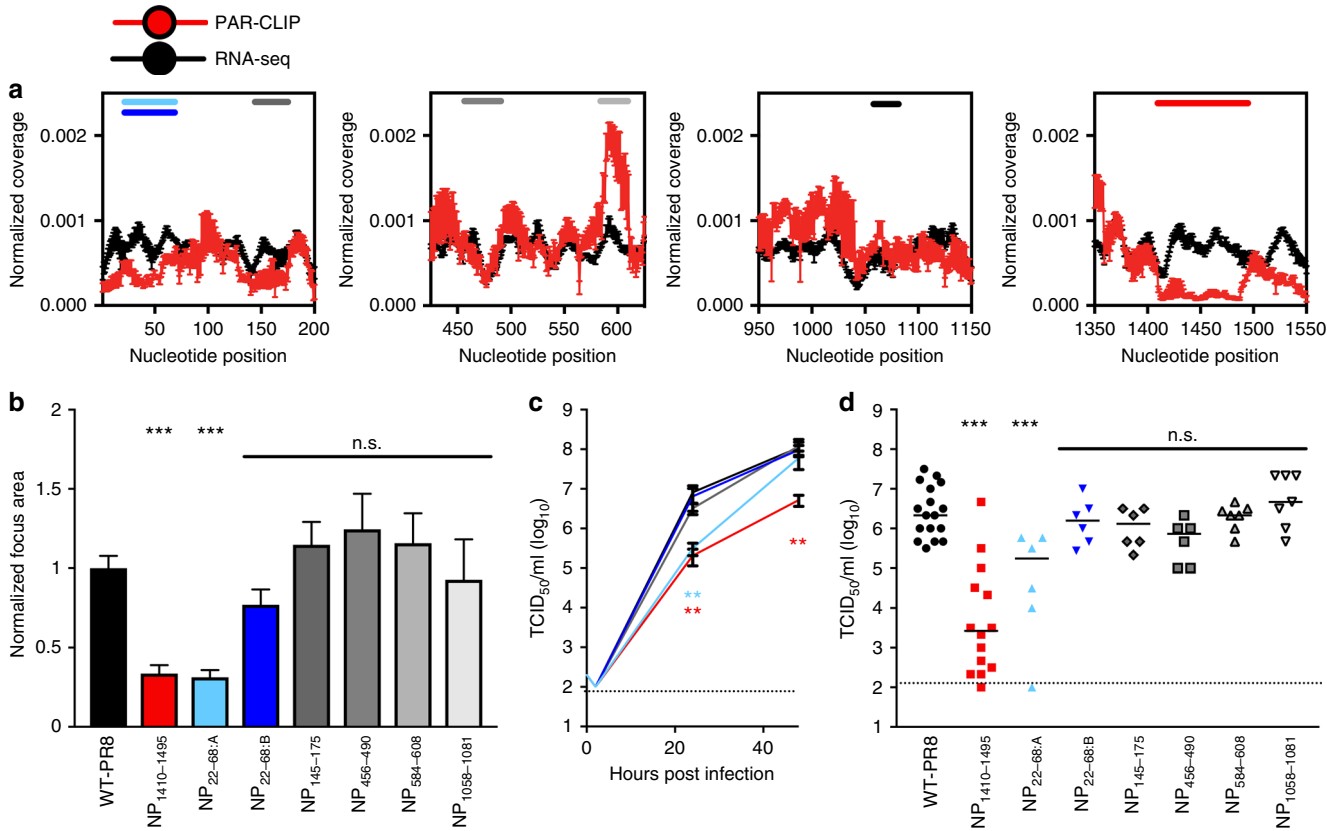

**Fig. 2** Manipulation of vRNA in low-NP-binding regions that disrupts predicted RNA structure attenuates virus replication. **a** Normalized coverage ± s.e.m. was determined for each nucleotide in the PAR-CLIP (red symbols) and RNA-seq (black symbols) libraries ($n = 4$ each). Six regions of interest (ROI) in segment 5 (NP) are highlighted, including two low-NP-binding regions ($NP_{1410-1495}$ and $NP_{22-68}$) and four intermediate NP-binding regions ($NP_{145-175}$, $NP_{456-490}$, $NP_{584-608}$, and $NP_{1058-1081}$). Each ROI is indicated with a colored bar. **b** Focus area of WT-PR8 and NP mutant IAV. MDCK cells were infected with serial dilutions of the indicated viruses and overlayed with infection media (M0.1B) containing 1% agarose and TPCK-trypsin. Seventy-two hours later cells were fixed, permeabilized, and stained for viral antigen (NP). Focus diameter was determined and normalized to WT-PR8 per experiment. Results are the average + s.e.m. of 3–5 experiments per virus (>60 foci each). **c** MDCK cells were infected with MOI = 0.001 of the indicated viruses and culture supernatant was collected at indicated time points then titered on MDCK cells. Results are the average ± s.e.m. $TCID_{50}$ $ml^{-1}$ of two experiments performed in duplicate. **d** C57BL/6J mice were inoculated with $10^{3}$ $TCID_{50}$ of the indicated viruses in 30 µL. Lungs were collected, homogenized, and titered on MDCK cells. Each dot is a single mouse and the line is the median. Dotted line in **c**, **d** represents the limit of detection. (***$P < 0.005$; **$P < 0.01$; n.s. not significant by one-way ANOVA with multiple comparisons correction (Kruskal–Wallis test))

synonymous mutations introduced within NP segment did not alter NP expression or its activity by all measurable outcomes.

We next assessed the virus particle to infectious unit ratio of WT and mutant PR8 viruses to evaluate potential defects in viral genome packaging. The infectious titer ($TCID_{50}$ $ml^{-1}$) of $NP_{1410-1495}$ and $NP_{22-68:A}$ mutant viruses was significantly lower ($P < 0.05$) compared to all other viruses when normalized to HA-units, indicating a greater production of non-infectious particles (Fig. 3d). The ability of WT and mutant viruses to package all eight genome segments was assessed using a population level measure of relative vRNA segment abundance in purified viral particles[22]. RNA from WT and mutant IAVs was subjected to RT-qPCR and the abundance of segments was normalized to segment 7 (Fig. 3e). Mutation of either the predicted pseudoknot ($NP_{1410-1495}$) or stem-loop structure ($NP_{22-68:A}$) resulted in aberrant genome constellation stoichiometry. Consistent with the viral replication assays, mutation of intermediate NP-bound regions or mutations designed to maintain the predicted 3′ stem-loop structure ($NP_{22-68:B}$) result in unaltered ratios of genome segments in all but one segment-specific instance (Fig. 3e and Supplementary Fig. 5a). These results suggest that RNA structure-destabilizing mutations, in regions of vRNA poorly bound by NP, impact replication at the stage of coordinated genome packaging.

Finally, we determined the co-expression of NP and matrix (M) proteins in singly infected cells (Fig. 3f, multiplicity of infection (MOI) of 0.05)[23]. In agreement with the RT-qPCR data, suggesting a packaging defect, a lower percentage of $NP_{1410-1495}$ and $NP_{22-68:A}$ mutant virus-infected cells co-expressed both proteins compared to cells infected with WT or other mutant viruses. Notably, high MOI infection with multiple defective particles simultaneously infecting the same cell is able to overcome this defect (Supplementary Fig. 5b).

**Additional low-NP-binding regions impact IAV replication.** Low-NP-binding regions in segments 1, 2, and 8 also were evaluated for their contribution to viral replication by mutagenesis designed to disrupt predicted secondary and tertiary RNA structures (Fig. 4a–c, Supplementary Table 1 and Supplementary Data 1). Nucleotide substitutions that alter the predicted RNA secondary structure but have no impact on the protein, i.e., synonymous structural mutations, in segment 1 ($PB2_{1823-1944:A-C}$), segment 2 ($PB1_{497-561:A}$) and segment 8 ($NS_{23-86:A}$) reduced the focus area of mutant viruses in MDCK cells (Fig. 4d–f). Control mutations, designed to maintain the predicted vRNA structure in the low-NP-binding regions on segment 2 and 8 ($PB1_{497-561:B}$, or $NS_{23-86:B}$), did not alter focus

**Table 1 Disruption of predicted RNA structures attenuates IAV**

| Virus | NP-bound | MFE | Effect of mutations | Effects on virus | | | |
| --- | --- | --- | --- | --- | --- | --- | --- |
| | | | | Focus size | Lung titer | Genome packaging | |
| | | | | | | $TCID_{50}$:HA ratio | Segment packaging |
| $NP_{22-68:A}$ | -- | −8.7 | ↓−$\Delta G$ | ↓↓ | ↓ | ↓↓ | ↓↓ |
| $NP_{22-68:B}$ | -- | −8.7 | No effect | = | = | = | = |
| $NP_{145-175}$ | = | −8.0 | ↓−$\Delta G$ | = | = | = | = |
| $NP_{456-490}$ | = | −3.2 | No effect | = | = | = | = |
| $NP_{584-608}$ | = | −0.0 | No effect | = | = | = | = |
| $NP_{1058-1081}$ | = | −0.0 | No effect | = | = | = | = |
| $NP_{1410-1495}$ | -- | −19.8 | Modified Pseudoknot | ↓↓ | ↓↓ | ↓↓ | ↓↓ |
| $PB2_{350-375}$ | -- | −0.0 | No effect | = | N.D. | = | = |
| $PB2_{1823-1944:A}$ | -- | −33.0 | Modified Pseudoknot | ↓ | ↓↓ | ↓↓ | = |
| $PB2_{1823-1944:B}$ | -- | −33.0 | Modified Pseudoknot | ↓ | N.D. | ↓ | = |
| $PB2_{1823-1944:C}$ | -- | −33.0 | Modified Pseudoknot | ↓ | N.D. | = | = |
| $PB2_{2213-2239}$ | = | −0.0 | No effect | = | N.D. | = | = |
| $PB1_{497-561:A}$ | -- | −12.9 | ↓−$\Delta G$ | ↓ | ↓ | ↓↓ | ↓↓ |
| $PB1_{497-561:B}$ | -- | −12.9 | No effect | = | N.D. | = | = |
| $PB1_{1828-1858}$ | = | −3.8 | No effect | = | N.D. | = | = |
| $PB1_{2032-2058}$ | = | −2.2 | No effect | = | N.D. | = | = |
| $NS_{23-86:A}$ | -- | −10.0 | Modified Pseudoknot | ↓↓ | ↓ | ↓↓ | ↓↓ |
| $NS_{23-86:B}$ | -- | −10.0 | No effect | = | N.D. | = | = |

For NP-binding, the following categories were considered: --, significantly lower than WT-PR8; =, equal to WT-PR8; ++, greater than WT-PR8. The regional stability was determined in WT-PR8 and mutant virus using Vienna RNAfold. All calculations were performed using the default settings without imposing structural constraints. Pseudoknot formation potential was determined using vsFold5 and default settings. Results for focus area, lung titer, $TCID_{50}$ ml$^{-1}$:HA ratio, relative segment packaging are summarized from previous figures (=, equivalent to WT-PR8; ↓, $P < 0.05$; ↓↓, $P < 0.01$)
*MFE* minimum free energy (expressed as $\Delta G$), *N.D.* not determined

size (Fig. 4e, f). Similar to the attenuation of structurally disrupted segment 5 viruses, the $PB2_{1823-1944:A-B}$, $PB1_{497-561:A}$ or $NS_{23-86:A}$ mutant viruses generated fewer $TCID_{50}$ particles per HA-unit, while the control mutations ($PB1_{497-561:B}$ or $NS_{23-86:B}$) did not (Fig. 4g–i). The $PB2_{1823-1944}$ mutant that showed the least attenuation in the focus-forming assay, $PB2_{1823-1944:C}$, had no discernable difference in $TCID_{50}$ per HA-unit. Segment 1 (PB2) and 2 (PB1) mutants did not display altered reporter activity (Supplementary Fig. 6a). In contrast to the impact of synonymous mutations in low-NP-binding regions, mutations in NP-bound regions in segment 1 ($PB2_{350-375}$ and $PB2_{2213-2239}$) and segment 2 ($PB1_{1828-1858}$ and $PB1_{2032-2058}$) did not alter focus size or $TCID_{50}$ per HA-unit (Fig. 4d, e, g, h, and Supplementary Fig. 5a). Specific manipulation of segment 8 vRNA sequence ($NS_{23-86:A}$) resulted in decreased packaging of segments 3, 4, 5, and 7 relative to segment 1 (Fig. 4j). Assessment of vRNA abundance in virus particles revealed that segment 2 ($PB1_{497-561:A}$) mutant viruses packaged reduced levels of segment 6 vRNA (NA) (Fig. 4j). The $PB2_{1823-1944:A}$ virus did not show altered segment abundance (Fig. 4k and Supplementary Fig. 6b). Finally, virus replication of the $PB2_{1823-1944:A}$, $PB1_{497-561:A}$ and $NS_{23-86:A}$ mutant viruses in mice was diminished 48 hpi compared to WT-PR8 (Fig. 4k). Together, these data suggest that vRNA sequences that are low in NP binding help to coordinate packaging of a full complement of eight vRNA segments and changes to the predicted RNA structures in these regions results in virus attenuation. To demonstrate that synonymous structural mutations affect the stability of the predicted RNA structure, we performed RNA thermal stability assays using RNA oligomers corresponding to the low-NP-binding region $PB1_{497-561}$[24]. The thermal profile of the RNA containing the predicted structure-destabilizing mutations ($PB1_{497-561:A}$) was different from that of the WT RNA oligomer and the RNA containing synonymous mutations that maintained the RNA structure ($PB1_{497-561:B}$) (Supplementary Fig. 4d). Moreover, the thermal melt profile for the destabilizing mutations ($PB1_{497-561:A}$) indicated a lower thermal stability. In contrast, the thermal profile of the $PB1_{497-561:B}$ RNA was similar to that of WT

RNA. Combined these data show that stable RNA structures are required for IAV replication and genome packaging.

**Mutation of low-NP-binding regions attenuates diverse IAV.** We evaluated the contribution of two low-NP-binding regions ($PB2_{1823-1944}$ and $NP_{1410-1495}$), identified in WT-PR8, to the replication of a North American avian IAV (A/shorebird/Delaware/22/2006 (IAV-H7N3)) and the 2009 pandemic H1N1 IAV (A/California/04/2009 (IAV-pH1N1)). The PB2 and NP gene-segment of IAV-PR8 and IAV-H7N3 are divergent and representative of mammalian and avian viruses, respectively (Fig. 5a, b, Supplementary Fig. 4). The NP gene-segment of IAV-pH1N1 is closely related to IAV-PR8 (Fig. 5a, Supplementary Fig. 4), while the PB2 gene-segment is more closely related to avian viruses (Fig. 5b, Supplementary Fig. 4). Isogenic IAV-H7N3 and IAV-pH1N1 viruses, bearing synonymous mutations in the $NP_{1410-1495}$ or $PB2_{1823-1944}$ regions, designed to disrupt the predicted local RNA structure, were evaluated for virus replication in vitro. Synonymous mutations designed to impact the predicted RNA secondary structure in these regions of IAV-H7N3 and IAV-pH1N1 attenuated viral replication, as measured by focus area (Fig. 5c, e). All mutant viruses displayed decreased $TCID_{50}$ per HA-unit (Fig. 5d, f). These results indicate that the RNA features, identified by PAR-CLIP in IAV-PR8, are important in genome packaging of diverse avian and human strains of IAV.

**Discussion**
We have identified the interaction landscape of IAV NP with viral RNA in the context of infected cells. Our findings indicate that binding of NP to viral RNA is restricted to an average of 12 nucleotides and the distance between two cross-linking sites is 25 nucleotides. These estimates agree with molecular models of NP–RNA interactions and the NP-binding footprint of a related orthomyxovirus[1,20,25,26]. Within this model, NP is excluded from consistent interaction with specific regions of vRNA and allows trans-interactions with either other genome segment vRNA or host and virus factors. The interaction between NP and the viral RNA was non-uniform and characterized by regions that were

consistently low or high in NP binding. These results are in agreement with recent findings from Lee et al.[27]. About half of the low-NP regions are predicted to form secondary and tertiary RNA structures, based on computational analysis, and mutations designed to alter the stability of these predicted RNA structures resulted in attenuated virus infection (summarized in Table 1). The presence of RNA structures in RNP complexes may explain the pleomorphic nature of RNPs[28]. Several, mostly shorter low-NP-binding regions were not predicted to form stable structures.

These regions may represent a portion of a larger but less stable structure or have a different yet unknown function during IAV replication. At present, the function of high-NP-binding vRNA regions is unknown. These regions may have a role in inter-segment NP-based interactions or be regions of high-NP density, enabled by position along the vRNP.

We examined NP–vRNA interactions at a late time point when a majority of viral RNA is distributed throughout the cytoplasm and thought to be within vRNP complexes[19,29]. In support of this,

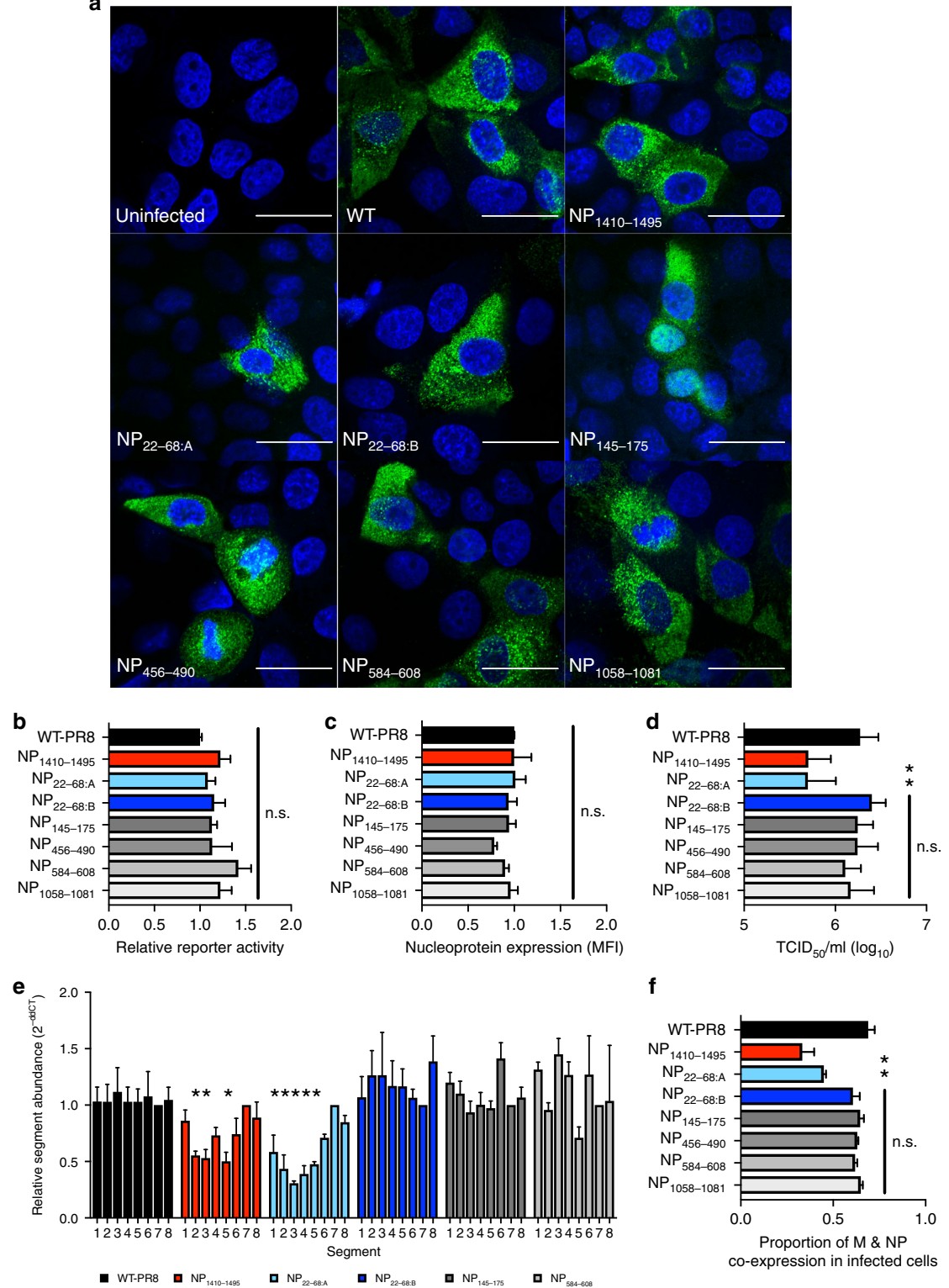

the great majority of PAR-CLIP reads were obtained from negative-sense vRNAs.

We found many low-NP-binding RNA regions that overlap with previously predicted packaging and bundling signals near segment termini (NP$_{22-68}$, NP$_{1410-1495}$, and NS$_{23-88}$)[5,30]. However, additional low-NP-binding regions were identified throughout the segment body and outside of traditional packaging signals[11,22,31]. Our findings agree with studies assessing structure-guided mutagenesis of hypothetical vRNA structures (segment 7) or biochemical analyses of vRNA (segment 8) that largely overlap with low-NP-binding regions assessed here[7,32]. Additionally, we identified specific nucleotides required for co-packaging of segment 2 (PB1) and segment 6 (NA), in a region recently implicated in directing reassortment outcomes and co-segregation of these segments in vitro[33,34]. Therefore, NP PAR-CLIP allowed us to directly identify potentially functional RNA regions in IAV genomes required for coordinated genome packaging.

As predicted RNA structures in some low-NP-binding regions are important for genome packaging, it suggests a requirement for inter-segment RNA interactions. Specific RNA interactions between genome segments have been demonstrated in vitro and ablation of these interactions lead to aberrant genome packaging; an outcome reversed by introduction of compensatory mutations in the interacting segment[12]. Many of the mutations that alter genome assembly also lead to an increase in number of defective viral particles[30]. The formation and nature of vRNA structures is likely contingent on the sequence composition and physical position of the nucleotides within a RNP complex. Additionally, manipulation of NP-bound regions, which may be unable to make inter-segment contacts due to NP-induced structural constraints on vRNA, did not impact virus replication. The presence of highly NP-bound vRNA regions may represent genomic regions that are spatially or structurally organized in a manner that allows improved protein binding, though we have not formally tested this hypothesis within this study.

IAVs can reassort and generate novel and potential pandemic strains. We tested if the regions identified in IAV-PR8 were required for replication of a divergent H7N3 virus. Synonymous changes designed to disrupt the predicted RNA structure in the same vRNA regions of two separate segments in both IAV-H7N3 and IAV-pH1N1 were attenuating and suggest conservations of these structural elements. Future examination of NP-binding of viral RNA in diverse strains of IAV is likely to identify overlapping and distinct high and low-NP-binding regions. These RNA features may be required for all viruses as coordinating packaging elements, but additional regions act as strain or lineage-specific packaging enhancers. Further, these strain or lineage-specific RNA features, required for genome packaging, may act as potential determinants of reassortment outcomes.

Prior experimental systems demonstrated that co-segregation of genome segments occurs during natural reassortment as well as lab-adaptation of virus strains to create new vaccines[34,35]. Finally, attenuation of genome packaging through silent, structural mutations has the potential to accelerate live attenuated vaccine production using native genome constellations to allow vaccination with all protein epitopes of a novel pandemic virus, without the need for master donor strains.

## Methods

**Cells.** Madin–Darby Canine Kidney (MDCK) cells were maintained in minimal essential medium (MEM) with 5% fetal bovine serum (FBS, Biowest), MEM-vitamins (Gibco), L-glutamine (Gibco), and penicillin–streptomycin (Gibco). Human embryonic kidney cells (293-T) were maintained in Opti-MEM (Life Technologies) with 10% FBS, L-glutamine, and penicillin–streptomycin. MDCK and 293T cells were a kind gift from Dr. Richard Webby at St. Jude Children's Research Hospital.

**Viruses.** Eight bidirectional pHW2000 plasmids containing cDNA for A/Puerto Rico/08/1934 (H1N1), A/California/04/2009 (H1N1), or A/shorebird/Delaware/22/2006 (H7N3) were previously described[35]. Viruses were generated by transfection of all eight plasmids into co-cultures of 293T and MDCK cells (1 µg per plasmid) with polyethylenimine (8 µg total). Wild-type (herein IAV-PR8 (WT-PR8), IAV-pH1N1 (WT-pH1N1) or IAV-H7N3 (WT-H7N3) and mutant viruses were generated in the same manner with the exception of substituting individual plasmids harboring mutagenized DNA for the single indicated wild-type plasmid. The next day, the transfection mixture was removed and replaced with Opti-MEM containing MEM-vitamins, L-glutamine and penicillin–streptomycin. Forty-eight hours post-transfection, an additional 1 ml of the same media containing 1 µg ml$^{-1}$ TPCK-Trypsin (Worthington) was added to the co-culture. Seventy-two hours after addition of TPCK-Trypsin, culture supernatant was collected and clarified by centrifugation (5 min, 350 × g). Viral stocks were generated by infection of MDCK cells in a T75 flask. Cells were washed once with PBS, and 200 µl of transfection supernatant was mixed with 25 ml of infection media (M0.1B) composed of MEM containing vitamins, L-glutamine, penicillin–streptomycin, 0.1% Bovine Serum Albumin (Gibco) and 1 µg ml$^{-1}$ TPCK-Trypsin for 48 h. Stocks were aliquoted and stored at −80 °C until use, and all studies were conducted with passage 1 stocks following verification of mutant sequence identity. All viruses were generated at least twice independently. Sequences for primers utilized during mutagenesis of IAV plasmids are available in Supplementary Table 2.

**IAV nucleoprotein PAR-CLIP and RNA-seq library generation.** To identify interactions between viral RNA and IAV nucleoprotein, we adapted the protocol for PAR-CLIP[21] coupled to next-generation sequencing to discern nucleotide resolution maps of protein–RNA interaction across the IAV genome. Confluent 293-T cells were infected at an MOI of 1 for 16 h in the presence of 100 µM 4-thiouridine (4-SU) and then cross-linked with ultraviolet light (310 nM, 500,000 µJ total energy, Boekel Scientific UV Crosslinker AH (115 V)). PAR-CLIP was performed essentially as described before[21] with the exception of antibodies used to immunoprecipitate NP–RNA complexes: monoclonal antibodies HB65 (ATCC) or MAb8258 (EMD Millipore). On-bead Calf Intestinal Alkaline Phosphatase (CIP, NEB) treatment was used to dephosphorylate RNA ends followed by 5′ end-labeling with $^{32}$P catalyzed by T4 Polynucleotide Kinase (PNK, NEB). Briefly, protein–RNA complexes were separated on 4–12% SDS-PAGE gels and transferred to nitrocellulose membranes. $^{32}$P-labeled RNA was identified by autoradiography, excised, and extracted following proteinase K (New England Biolabs (NEB)) digestion. To account for potential bias introduced during adaptor ligation and sequencing, we also generated RNA-seq libraries using RNA extracted from

**Fig. 3** Attenuating mutations in segment 5 impact coordinated genome packaging. **a** Confocal microscopy images depicting the localization of nucleoprotein (NP) 8 h post infection (hpi). MDCK cells were infected with the indicated virus (white text), and NP was identified using the MAb HB65 and Alexa-488-conjugated goat-anti-mouse secondary antibody by immunofluorescence. Cell nuclei are stained with DAPI. Panels depict the merged image of DAPI and NP staining. Fields are representative of two independent experiments. Scale bar represents 25 µm. **b** Dual-luciferase reporter assay to assess viral transcription and genome replication, each combination of plasmids was assessed 3–5 times with corresponding the WT-PR8 combination. (n.s., not significant by one-way ANOVA with multiple comparisons correction (Kruskal–Wallis)). **c** Mean fluorescence intensity (MFI) of NP in virally infected MDCK cells (MOI = 0.05, experiments were performed twice in duplicate) as revealed by flow cytometry. (n.s., not significant by one-way ANOVA with multiple comparisons correction (Kruskal–Wallis)). **d** Viral titer (TCID$_{50}$ ml$^{-1}$) of 4 HA-units of WT-PR8 and mutant viruses. Results are the average + s.e.m. of three viral titrations and two HA assays per infectious virus titration. (*$P < 0.05$; n.s., not significant by unpaired t test). **e** Relative abundance of genome segments in purified WT or mutant viruses. All segments were compared to segment 7 (M) vRNA and normalized to the average of WT-PR8 values using the $2^{-ddCt}$ method[22]. Bars represent the mean of 3–6 independent virus preparations + s.e.m. (*$P < 0.05$; n.s., not significant by one-way ANOVA with multiple comparisons correction (Kruskal–Wallis)) **f** Proportion of infected cells co-expressing of matrix (M) and NP proteins in singly infected MDCK cells (MOI = 0.05) 16 hpi as revealed by flow cytometry. The average percentage of co-expression was calculated from two experiments performed in duplicate. (n.s., not significant by one-way ANOVA with multiple comparisons correction (Kruskal–Wallis))

uncross-linked cell lysates of influenza infected 293T cells from which RNA was isolated by TRIzol (Invitrogen) extraction, then 10 µg RNA was fragmented by $Mg2^+$ at 95 °C for 12 min (NEB, Magnesium RNA Fragmentation Module). $^{32}P$-labeling of this RNA was performed in solution with T4 PNK (NEB), and size-selection of 10–100 nt fragments by Urea-Page (15%) was performed. In both cases, total RNA was precipitated with ethanol, and used to prepare Illumina sequencing libraries. 3′-OH RNA previously prepared by CIP treatment was ligated to a 3′ adaptor using T4 RNA Ligase 2, truncated K227Q (NEB), according to manu-facturer's instructions. The 5′ adaptor sequence contains NNN-degenerate nucleotides in addition to a sequence used for demultiplexing to facilitate

collapsing of redundantly generated PCR products and ascertain the frequency of individual NP–RNA interaction events. This adaptor was ligated to the 5′-end of RNA using T4 RNA Ligase 1 (NEB). Following isolation of adaptor ligated and radiolabeled RNA, all subsequent library generation steps were identical. Inde-pendent PAR-CLIP or RNA-seq libraries were pooled and cDNA synthesized using a primer complementary to the 3′ adaptor (SuperScript III, Invitrogen). Libraries were amplified by PCR for 9–15 cycles using Phusion DNA polymerase (Phusion HF Mastermix, Thermo) and primers annealing to the 5′ and 3′ termini of the DNA that enable flow-cell binding. Size selection of libraries was conducted by extraction of amplicons from 6% urea gel electrophoresis, and then libraries were

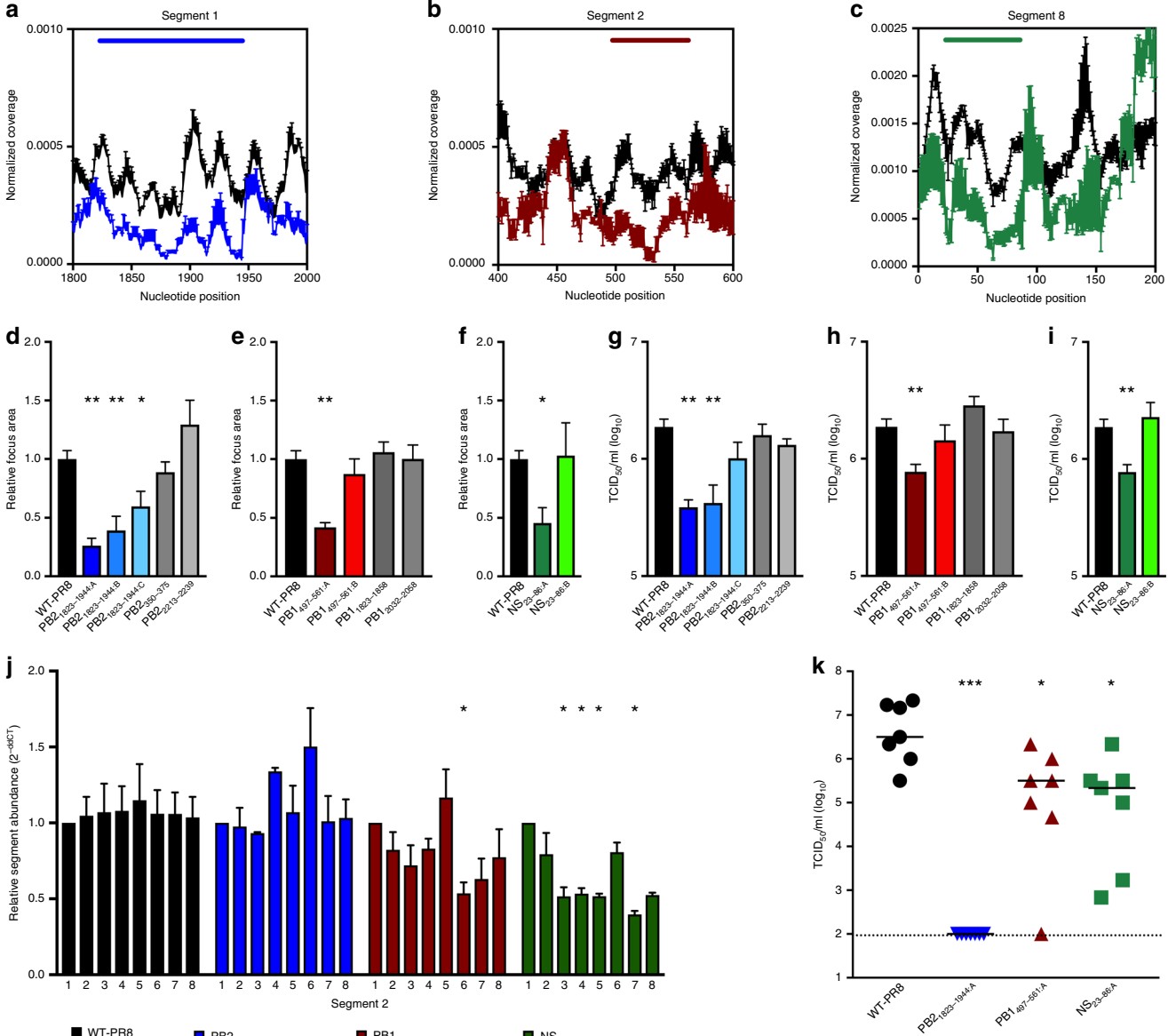

**Fig. 4** Synonymous structural mutations in low-NP-binding regions in Segments 1, 2, and 8 attenuated virus replication and genome packaging. **a–c** Segment 1 (**a**, blue line), 2 (**b**, red line), and 8 (**c**, green line) low-NP-binding regions defined by PAR-CLIP analysis. Normalized coverage ± s.e.m. was determined for each nucleotide in the PAR-CLIP (blue, red or green symbols) and RNA-seq (black symbols) libraries (n = 4 each). **d–f** Relative focus area of WT-PR8 and segment 1, 2, and 8 mutant viruses. Results are the average focus area + s.e.m. of 3 experiments per virus (>60 foci each). Statistical significance was determined by one-way ANOVA with a multiple comparisons correction (Kruskal–Wallis test). **P < 0.01; *P < 0.05. **g–i** Viral titer ($TCID_{50}$ ml$^{-1}$) of 4 HA-units of WT-PR8 and mutant virus. Results are the average of three viral titrations and two HA assays per infectious virus titration ($TCID_{50}$ assay and HA titration experiments, mean + s.e.m.). Statistical significance was determined by unpaired t test. **P < 0.01; *P < 0.05. **j** Relative abundance of genome segments in purified WT or mutant viruses. All segments were compared to segment 1 (PB2) vRNA and normalized to the average of WT-PR8 values using the $2^{-ddCt}$ method[22]. Bars represent the mean of 3–6 independent virus preparations + s.e.m. and statistical significance was determined by one-way ANOVA with a multiple comparisons correction (Kruskal–Wallis test). *P < 0.05. **k** C57BL/6J mice were inoculated with $10^3$ $TCID_{50}$ in 30 µL. Lungs were collected 48 hpi, homogenized, and titered. Each dot is a single mouse and the line is the median. Dotted line in **k** represents the limit of detection and statistical significance was determined by one-way ANOVA with a multiple comparisons correction (Kruskal–Wallis test). (***P < 0.005, *P < 0.05)

precipitated. Molarity of libraries was determined by qPCR (NEBNext, New England Biosciences) and size by Bioanalyzer (Agilent). Illumina sequencing on the HiSeq 2500 instrument (1 × 50 bp reads) was performed by the Genome Technology Access Center at Washington University in St. Louis.

**Next-generation sequencing**. All NGS data were analyzed on the Washington University in Saint Louis School of Medicine's McDonnell Genome Institute (MGI) cluster using publically available analysis programs (FastX toolkit, Bowtie, and SAMtools) and in-house scripts[21]. PAR-CLIP and RNA-seq data were generated from four independent experiments. For PAR-CLIP and RNA-seq libraries we used an analysis pipeline that collapsed unique barcoded reads, removed adaptors, and

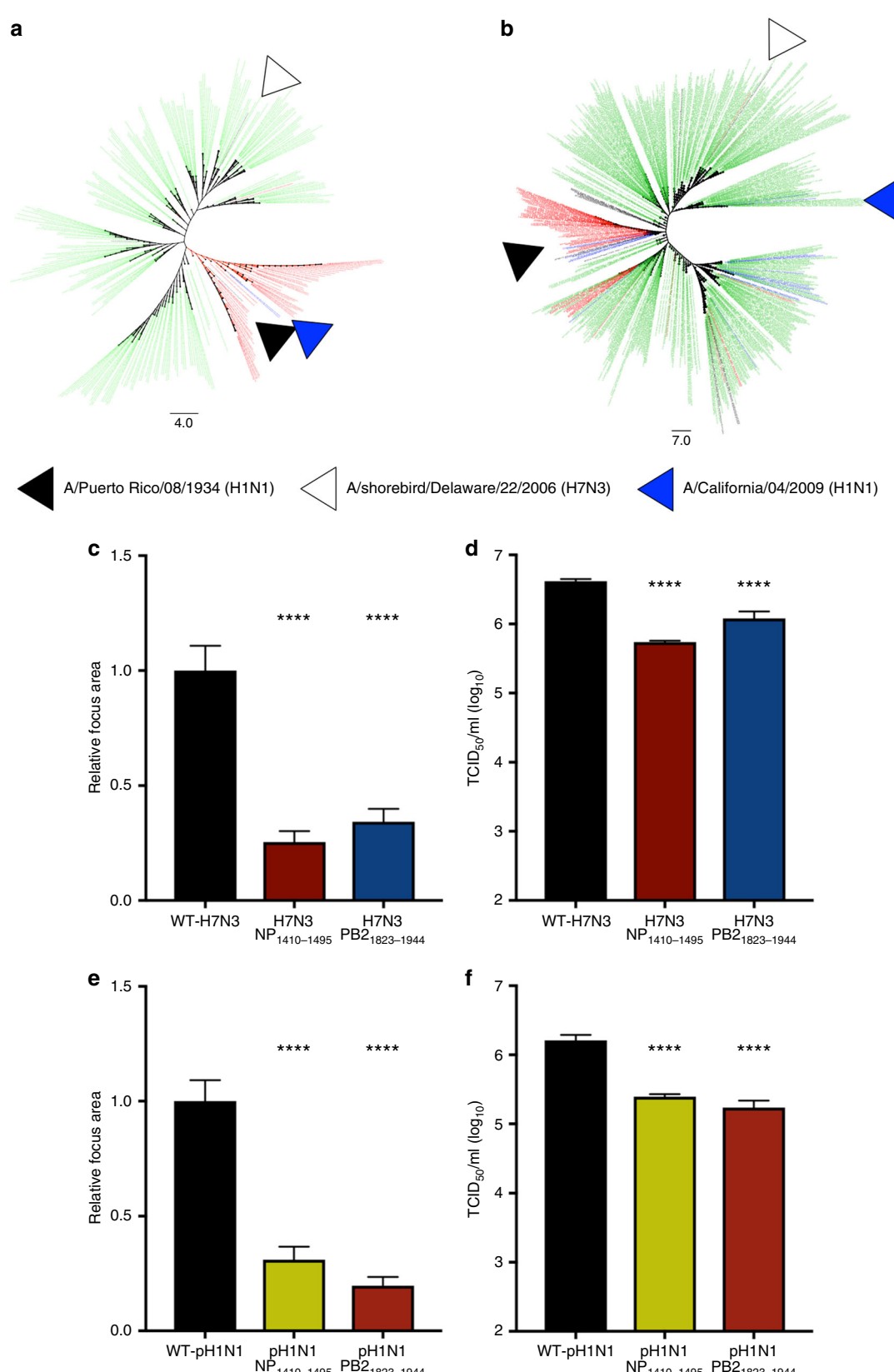

aligned them to the viral genome. Strandedness was determined post-alignment and all subsequent analysis was performed on negative-sense viral genomic RNA (vRNA). For reads that mapped to the viral RNAs, we normalized the number of reads per nucleotide to the total number of reads per genome segment to yield a normalized coverage ratio for both PAR-CLIP and RNA-seq libraries. We then compared the normalized coverage for each preparation at nucleotide resolution using an unpaired $t$ test with False Discovery Rate (FDR) correction (Benjamini–Hochberg)[36]. vRNA nucleotides with a FDR minima $Q < 0.01$ and fold-change > 3 were identified and these regions were extended to the final nucleotide of each regionm, where $Q \leq 0.1$. These areas represent low-NP or high-NP-binding regions of interest (ROI). ROI $\geq 18$ nt were subjected to RNA structure analysis with mFold, RNAfold, and vsFold5 to determine computationally the theoretical minimal free energy ($\Delta G$) of each region as well as the potential secondary or tertiary RNA structure formation[37,38]. Nucleotide composition of ROIs was determined by calculating the percent of A, G, C, or U in each region and comparing them to the percent of each nucleotide in the IAV genome. PAR-CLIP data sets were compared to RNA-seq data sets at each nucleotide position on a per-segment basis and positions using unpaired $t$ test with FDR correction as above with a threshold of $Q = 0.05$. Sites of statistically enriched transition in PAR-CLIP data sets were tabulated and the sequence distance between these positions was calculated. Transition events with a step-size of 1, i.e. sequential nucleotides both with high transition rate, and were excluded from analysis because they likely represent single NP–RNA interactions of adjacent 4-SU nucleotides in vRNA. Transition rate for four PAR-CLIP libraries were averaged prior to distance calculation.

**Generation of IAV containing altered RNA structures**. Mutant selection was performed after analysis of open reading frames to assess potential nucleotide degeneracy while retaining coding sequence. Subsequently, we selected five underrepresented ROIs in four segments (1, 2, 5, and 8) for functional assessment of manipulating predicted RNA structures on viral replication. We also identified four regions in segment 5, amenable to extensive silent mutagenesis that were either highly bound or represented at the same frequency in PAR-CLIP and RNA-seq data sets. We introduced between 2 and 7 synonymous mutations simultaneously in silico to the identified regions and reassessed structural stability or pseudoknot formation using the same structure prediction programs. We selected variant codon combinations that would disrupt or maintain the predicted vRNA structure but not change the encoded amino acid, alter codon usage, or disrupt alternative reading frames or splicing events. Once predicted destabilizing mutations were identified, mutant viruses were generated. Mutations were introduced into pHW2000 bidirectional plasmids by inverse PCR with primers (see Supplementary Table 2) including selected mutations and unique ligation sites. PCR products were gel extracted and digested with either BsmbI (New England Biolabs, (NEB)) or AarI (Thermo Fisher Scientific) restriction enzymes and DpnI (NEB) to remove residual parent plasmid. Digested PCR products were PCR purified and ligated using Instant Sticky End Ligase (NEB). Ligation products were transformed into E. coli, plated on selective LB Agar. Colonies were selected and grown overnight in LB broth and selective antibiotics. All plasmids were grown in 200 ml LB Broth prior to preparation by Qiagen HiSpeed Endotoxin-Free MaxiPrep and verified by Sanger Sequencing (Genewiz).

**Influenza A virus focus-forming assay**. Six-well plates of MDCK cells were plated and inoculated with 1 ml of 10-fold serially diluted virus stock for 1 h in M0.1B. After 1 h, the inoculum was removed, and replaced with an overlay of 1% Low Melting Point Agarose in MEM supplemented with 1 μg ml⁻¹ TPCK-Trypsin. Seventy-two hours post infection cells were fixed in 5% formaldehyde, permeabilized with 0.1% saponin in HBSS, and stained for NP protein with 1 μg ml⁻¹ biotinylated anti-NP (mAB8258b, Millipore). Foci were visualized by addition of HRP-conjugated streptavidin and TruBlue streptavidin substrate and imaged on a Biospot reader (Cellular Technology Limited). Individual foci were counted and area was calculated using the Analyze Particles extension within Fiji[39]. The area of

the foci formed by mutant viruses was normalized to the average of WT-PR8 foci per experiment. A minimum of 60 foci was analyzed per virus.

**Confocal microscopy**. We infected 293T cells in the presence or absence (mock) of 4-SU with WT-PR8 (MOI of 1) for 16 h. Briefly, 293T cells were seeded on glass coverslips coated with poly-D lysine overnight. Cells were infected with WT-PR8 for 16 h then fixed with 4% methanol-free PFA in PBS (pH 7.4), washed with PBS, permeabilized with saponin, then NP staining was performed using anti-NP MAb HB65 (1 μg ml⁻¹) and goat-anti-mouse Alexa-488 (Invitrogen). Nuclei were counterstained with DAPI. Coverslips were mounted to slides with ProLong® Diamond Antifade Mountant (Molecular Probes). To determine if viral protein production was impacted by synonymous mutations, we utilized confocal microscopy to visualize the distribution of NP during infection as a marker of viral protein production and a proxy for trafficking of vRNPs. MDCK cells were infected with WT-PR8 and indicated mutant viruses (MOI of 0.2) for 8 h, then fixed and permeabilized as before. NP staining was performed as above using anti-IAV-NP MAb HB65 and Goat-anti-Mouse Alexa-488 (Invitrogen). Slides were imaged with Zeiss LSM 880 Confocal Laser Scanning Microscopy with Airyscan and analyzed with Zeiss Zen Black Software performed within the Molecular Microbiology Imaging Facility at Washington University in Saint Louis.

**Influenza A virus reporter assay**. To assess the impact of silent mutations on polymerase complex activity, pHW2000 plasmids encoding WT or mutant PB2, PB1, PA, and NP were utilized in a dual-luciferase reporter assay as previously described[35]. A vRNA-like firefly luciferase reporter plasmid and a Renilla luciferase expression plasmid also were included. Briefly, 293T cells were seeded into 24-well plates and transfected with equal amounts of all six plasmids (500 ng DNA total) in Opti-MEM containing TransIT-LT1 (Mirus). Cells were maintained at 37 ℃ for the duration of the experiment. Forty-eight hours later, cells were lysed for analysis of luciferase activities (Promega). Each combination of polymerase proteins (set of plasmids) was examined in duplicate. The relative light units (RLU) of firefly luciferase activity were normalized to the RLU for Renilla luciferase activity within the same sample to account for differences in transfection efficiency between wells and experiments. The activity of each plasmid set containing a mutant segment was normalized to the activity of WT-PR8.

**Flow cytometry**. To determine the amount of NP generated at a fixed time during infection, we infected $2 \times 10^5$ MDCK cells (MOI of 0.2). Eight hours post-infection, cells were collected and fixed. Intracellular staining of IAV antigen was performed as above using an anti-IAV NP primary antibody (HB65) (1 μg ml⁻¹) and an Alexa-488-conjugate goat-anti-mouse secondary antibody (Invitrogen). The mean fluorescence intensity (MFI) of NP⁺ cells was plotted and calculated from two experiments performed in duplicate. Co-expression of NP and M in MDCK cells was determined 16 hpi with an MOI of 5 and 0.05. Infected cells were fixed and stained as above with the additional staining step of utilizing mouse-anti-M primary antibody (M2-1C6) (2 μg ml⁻¹) and goat-anti-mouse Alexa-647 secondary. A second intracellular staining step utilizing biotinylated mouse-anti-NP (mAb8258b) (1 μg ml⁻¹) and Alexa-488 conjugated streptavidin was then performed. The frequency of co-expression was calculated by determining the number of cells expressing NP or M as well as those expressing both. The percentage of infected cells co-expressing NP and M was calculated by dividing co-expressing cells by all cells expressing one or more viral protein as previously described[23]. Analysis of viral proteins was determined using a flow cytometer and FlowJo software (Tree Star).

**Multi-step replication of influenza A virus**. MDCK cells ($2 \times 10^5$) were seeded in 24-well plates and inoculated the next day with 200 TCID₅₀ of IAV. Cells were washed once with PBS before addition of inoculum in M0.1B and incubated for 1 h at 37 ℃. Subsequently, the cells were washed twice with PBS and 1 ml of M0.1B supplemented with 1 μg ml⁻¹ TPCK-Trypsin was added to each well. Culture

**Fig. 5** vRNA regions required for PR8 replication are required for replication of contemporary avian and human IAV. **a** Phylogenetic analysis of segment 5. WT-PR8, WT-H7N3, or WT-pH1N1 IAV are indicated by corresponding, labeled arrows. Phylogenies were created from randomly sampled full-length segment 5 sequences downloaded from NCBI IVR. Alignment performed in MEGA (version 7) using MUSCLE. Phylogenetic trees created using the Maximum Parsimony method included in MEGA. Phylogenies were visualized in FigTree and manually annotated. Green shading represents segment sequences derived from avian viruses; red represents human viruses; and blue represents swine viruses. **b** Phylogenetic analysis of segment 2 performed as in **a**. **c** Relative focus area of WT-H7N3 and mutant viruses + s.e.m. The results are the average of 3 independent experiments and >60 foci per virus. Statistical significance was determined by one-way ANOVA with multiple comparisons correction (Kruskal–Wallis test). ****$P < 0.001$. **d** Viral titer (TCID₅₀ ml⁻¹) of 4 HA-units of WT-H7N3 and mutant virus. Results are the average of three viral titrations and two HA assays per infectious virus titration (TCID₅₀ assay and HA titration experiments, mean + s.e.m.). Statistical significance was determined by unpaired $t$ test. ****$P < 0.001$. **e** Relative focus area of WT-pH1N1 and mutant viruses + s.e.m. The results are the average of 3 independent experiments and >60 foci per virus. Statistical significance was determined by one-way ANOVA with multiple comparison corrections (Kruskal–Wallis test). ****$P < 0.001$. **f** Viral titer (TCID₅₀ ml⁻¹) of 4 HA-units of WT and mutant virus. Results are the average of three viral titrations and two HA assays per infectious virus titration (TCID₅₀ assay and HA titration experiments, mean + s.e.m.). Statistical significance was determined by unpaired $t$ test. **** $P < 0.001$

supernatants were collected at 24 and 48 hpi and the amount of infectious virus was quantified by titration on MDCK cells.

**Titration of influenza A virus (TCID$_{50}$)**. Confluent monolayers of MDCK cells were grown overnight in 96-well tissue culture plates. The next day, the cells were washed with PBS and inoculated with 10-fold serial dilutions of culture supernatant or lung homogenate for 1 h in M0.1B at 37 °C. After 1 h, the inoculum was removed and replaced with M0.1B supplemented with 1 µg ml$^{-1}$ TPCK-Trypsin and incubated for 72 h. Presence of virus was determined by hemagglutination assay using 0.5% turkey red blood cells. TCID$_{50}$ was determined by the Reed–Muench method[40].

**Infection of mice with influenza viruses**. Male C57BL/6 J mice (5–6 weeks of age) were bred in-house in a barrier facility at Washington University School of Medicine, St. Louis, MO or purchased from Jackson Laboratories. Mice received food and water ad libitum and all experiments were conducted in accordance with rules of the Institutional Animal Care and Use Committee. Mice were anesthetized with isoflurane in an airflow chamber and then inoculated intranasally with 30 µL of sterile PBS containing 1000 TCID$_{50}$ of WT-PR8 or the indicated mutant virus. Forty-eight hours post-infection, lungs were collected and homogenized in 1 ml M0.1B, cleared by centrifugation at 1200 × g for 5 min, then stored in aliquots at −80 °C. Viral titers from lung homogenates were determined by TCID$_{50}$ assay.

**Segment abundance RT-qPCR**. MDCK cell derived stocks of WT-PR8 or mutant viruses were clarified by centrifugation at 1200 × g for 5 min, passed through a 0.22 µM filter, and pelleted on 30% sucrose cushion by ultracentrifugation (Beckman SW32ti swinging bucket rotor, 27 K RPM, 4 °C for 90 min). Pelleted virus particles were directly resuspended in 350 µl TRK lysis buffer, and RNA purified immediately and eluted in 30 µl DEPC H$_2$O (Total RNA Kit I, Omega). cDNA was synthesized from 5 µl of RNA with SSIII Reverse transcriptase (Invitrogen) and a vRNA specific primer[22]. Total cDNA was diluted 1:10,000 and used to quantify each of the 8 genome segments by SYBR Green qPCR (PowerUp$^{TM}$ SYBR$^{\circledR}$ Green Master Mix) with primer pairs previously published[22]. Relative abundance of each genome segment was calculated as before, except with normalization to Segment 1 or 7 depending on the virus assessed.

**RNA thermal denaturation experiments**. The RNA oligomers corresponding to PB1$_{497-561}$, PB1$_{497-561:A}$, and PB1$_{497-561:B}$ (Supplementary Table 2) were synthesized commercially (Integrated DNA Technologies). RNA was diluted in buffer (100 mM NaCl and 50 mM KCl) to a final concentration of 10 µM, incubated at 95 °C for 5 min and flash cooled on ice to induce folding. Unfolding of RNA secondary structure was monitored by CD spectroscopy at 210 and 260 nm on a Chirascan CD spectrometer (Applied Photophysics)[41]. RNA was heated from 5 to 95 °C at a rate of 1 °C/min and CD readings collected every 2 °C. A linear, least-squares program was used to fit the transition region of thermal melting curves. By definition, the melting temperature of nucleic acid is the temperature at which 50% of all nucleic acid becomes single-stranded for a single cooperative transition[24]. The ellipticity at which half of the RNA is denatured was calculated using the equation shown below and modified from (42):

$$\mathrm{Fd} = (\theta\mathrm{n} - \theta\mathrm{obs})/(\theta\mathrm{n} - \theta\mathrm{d})$$

where, Fd is the fraction of denatured nucleic acid, $\theta$n the minimum ellipticity at 210 nm or the maximum ellipticity at 260 nm, $\theta$d the maximum ellipticity at 210 nm or the minimum ellipticity at 260 nm, and $\theta$obs the observed ellipticity at Fd. The ellipticity, $\theta$obs, when Fd = 0.5 was calculated and used to calculate Tm from the linear fit of the transition region. For comparisons between different RNA samples, the first derivative of the ellipticity, $\theta$obs, the change in molar ellipticity as a function of temperature (d$\theta$/d$T$), was calculated and plotted.

**Statistical analysis**. All statistical analyses were performed using GraphPad Prism 7.0. For comparison of PAR-CLIP and RNA-seq data sets, we used multiple unpaired $t$ tests with the Benjamini–Hochberg correction to identify areas in which these sequencing preparations were statistically different from each other ($Q < 0.1$ and $Q < 0.01$). Transition-distance was determined using FDR ($Q < 0.05$). For analysis of the focus-forming assay, luciferase assay, and lung viral titers, we used one-way ANOVA with multiple comparison corrections (Kruskal–Wallis test). TCID$_{50}$ per HA-unit was analyzed using an unpaired $t$ test.

**Data availability**. The unprocessed PAR-CLIP and RNA-seq files have been uploaded to the NCBI Sequence Read Archive (SRA, SUB3042681). All other data supporting the findings of this study are available within the article and its Supplementary Information files, or are available from the authors upon request.

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

## Acknowledgements

We thank the following colleagues for comments on experiments comprising this manuscript: Michael Diamond, MD, David Wang, PhD, Daisy W. Leung, PhD, and Deborah Lenschow, MD and Anshu Gounder all of Washington University in Saint Louis. Additionally, Traci Bricker and Christopher Edwards provided technical support. Chris Brooke provided anti-M1 hybridoma, M2-1C6. NIGMS training grant 5T32GM007067 and NIAID ID Training Grant 2T32AI007172, and the Victoria Fraser Infectious Disease Research Fellowship provided funding to GW. This study was funded in part by the National Institute of Health (NIH) grants P01AI120943 (G.K.A) and R01-AI118938 (A.C.M.B.).

## Author contributions

G.D.W. and A.C.M.B. conceived and performed the experiments, analyzed the data, and wrote the manuscript. S.B.K. and D.T. provided technical and conceptual assistance in performing PAR-CLIP experiments. K.M.W. provided bioinformatics analysis support. G.K.A. assisted with the analysis and conception of certain experiments concerning RNA structure. P.J.K. and G.K.A. performed the RNA thermostability assays.

## Additional information

**Competing interests:** The authors declare no competing financial interests.

