## [Peer Review File · Nature Communications]

Reviewers' comments:

Reviewer #1 (Remarks to the Author):

Several cryo-EM structures of the influenza A virus (IAV) viral ribonucleoproteins (vRNPs) showed that nucleoprotein (NP) forms a helical scaffold but revealed nothing about the structure of the viral RNA segments (vRNAs) as they were not visible in the density maps. The vRNA structure likely has a profound impact on packaging and genetic reassortment of IAVs and studies addressing the vRNA structure within intact viral particles or purified vRNPs are welcome. Graham et al. used PAR-CLIP to study the genome-wide association of NP with vRNAs in infected cells. Their work represents a very important contribution to the field. There are however a few points that need to be addressed before publication.

Major points

1) The authors conclude that RNA structure in low-NP regions is required for vRNA packaging (e.g lines 453-454). While the authors clearly show that mutations in the low NP-regions have an effect on viral replication, the implication of vRNA structure on packaging is suggested, but not convincingly demonstrated. The authors should add a few additional mutants to strengthen this point and modulate their conclusions.

1.1. Indeed, there is not experimental structural analysis of vRNAs, only secondary structure predictions that were performed on the regions of interest (ROI) (lines 283-287). It is well known that RNA structure predictions are very sensitive to the length of the sequence included in the analysis. Adding or removing a few nucleotides at either end of the ROI would probably significantly affect the analysis. Also these ROI might form long-range interactions with other parts of the vRNA segment that are completely ignored.

1.2. Only in one case are mutations predicted to affect RNA structure compared to mutations in the same region predicted to have no effect on the structure (mutants NP(22-68:A) and NP(22-68:B)). For all the other low-NP regions analyzed, one cannot exclude that mutations that would not affect the vRNA structure would have the same effect as the mutations that affect vRNA structure. Thus, the authors should analyze the effect of mutations not predicted to affect structure of (two or three) other low-NP binding regions and show that they don't affect replication and packaging.

1.3. The three viruses with mutations in the PB2(1823-19944) region demonstrate that mutations within low-NP regions can affect viral replication without having any negative effect on packaging (lines 184-188, Fig 4f and Fig 6b,c) in contradiction with their conclusion (lines 28-29, 188-191, 453-454). I appreciate the honesty of the authors who included this data in their manuscript even though it doesn't completely fit their main message, but they shouldn't forget it and modulate their conclusion.

2) The authors conclude that mutations in NP-bound regions have no effect (e.g. lines 29-30). This conclusion is based on four mutants that are all in the NP segment (Fig 2). Mutations of NP-bound regions in other segment(s) would greatly reinforce this conclusion. This is an important point, as this would also reinforce the idea that NP-free and NP-bound regions have different functions.

3) In order to conclude that 4-SU has no effect on viral replication, the authors should not just show viral titers at 24 hpi (Fig 1b) but replication curves.

4) Analysis of A-to-G mutations:

4.1. These should be A-to-G mutations in the (+) strand, right? The X-linked 4-SU in the (-) (genomic) strand is converted as a G in the (+) strand during reverse transcription. Lines 291-293 are confusing (false?).

4.2. Isn't it possible to draw more information from this data than the spacing between the X-linking site? For instance, it would be interesting to know whether this spacing is the same in the low- and high-NP regions. A map, profile (similar to Supplementary Fig 3) or table of the X-linked site could also be very useful.

Minor points:

1) Introduction, lines 45-45. The paper by Klump et al. (1997) EMBO J15, 1248 should be cited.

2) Legend to Supplementary Fig 2 should be more detailed

3) Supplementary Fig 3 should be much larger, so that readers can retrieve precise information from it

4) line 188: Supplementary Fig 5 b-c should read Supplementary Fig 6 b-c

5) The predicted structural effects of the mutations introduced in H7N3 and H1N1 (Fig 5c-d) should be shown in a supplementary figure.

6) Line 241: the wavelength and the apparatus used for X-linking should be indicated.

7) Line 254: More details about the preparation of the Illumina sequencing libraries must be provided (amount of RNA, ligation conditions, ...)

Reviewer #2 (Remarks to the Author):

In this manuscript by Williams and colleagues use PAR-CLIP to assess the interaction of the influenza virus nucleoprotein (NP) with viral genomic RNA (vRNA) in human cells infected with influenza A/PR/8/34 (H1N1) virus. They find that NP binds to vRNA non-uniformly and without sequence specificity. They identify regions of low NP occupancy as well as regions enriched in NP. Mutation of low NP binding regions resulted in virus attenuation and in most cases this attenuation correlated with a defect in genome packaging. On the other hand, mutations in an NP-bound region had no effect. They extend these studies to a pH1N1 and an avian H7N3 virus by showing that mutations in regions corresponding to the low NP binding regions in PR8 attenuate the replication of these viruses.

Overall, this is the first study that addresses the genome-wide association of NP in an influenza virus. The major finding is that NP is non-uniformly distributed along the vRNAs. Although, this has been suggested by previous studies, this paper provides convincing experimental evidence for this model. Overall, the results are clearly presented and for most parts support the conclusions. The manuscript would benefit from addressing the following points.

1. Page 3, lines 35-38. It is an over-interpretation that refs 1 and 2 solve the structure of the protein components within vRNP. These studies provide little information on the overall structure of the polymerase and propose two contradictory models for the arrangement of NP.
2. Fig. 1b. The NP signal is hardly visible in the IF panel and this figure adds little to the manuscript. Either improve or delete.
3. Page 6, lines 95-100 and Sup. Tab. 1. The labelling in this table and the interpretation of the data is confusing. Bold does not seem to be used consistently and the numbers of low and high NP binding regions in the text and table do not match. Should 'Low (nt)' read 'High (nt)' under Segment 4 on the right? Lines 98-99. 'four high-NP regions that bound RNA were overrepresented ...' is unclear.
4. Page 6, lines 102-104. 'Analysis of low-NP binding regions revealed that RNA secondary structures might form in the absence of NP binding in some of these regions.' This statement is not supported by data or references.
5. Page 7, lines 112-116. NP region 586-608 was included in the analysis as representative of highly bound vRNA. However, this region is not identified as high NP region in Sup. Tab. 1.
6. Page 7, lines 116-119 and Sup. Fig. 4. There are no experimental data presented to support that the proposed secondary structures form and the introduced mutations disrupt these structures.
7. Page 8, lines 140-143. NP cannot transcribe and replicate.
8. Page 9, line 188. 'Supplementary Fig 5 b-c' should read 'Supplementary Fig 6 b-c'.
9. Fig. 4. It is intriguing that the PB2 segment mutations result in the largest attenuation of the virus but there is no detectable defect in vRNA packaging. There is no attempt to determine the mechanism of attenuation for these mutants. Given that out of the three regions analysed only two are important for coordinating packaging the authors should soften their conclusion (lines 188-191).
10. Fig. 5. Although the introduced mutations result virus attenuation, there is no attempt to link this to a defect in genome packaging. It would be useful to see data on this as well as PAR-CLIP data and an alignment of the relevant RNA regions between PR8, pH1N1 and H7N3 to be able to assess conservation. Secondary structure predictions should also be shown if different from those shown for PR8 in Sup. Fig. 4.
11. Page 33 and line 650-656. Legends for panels e and f are switched around.
12. Page 20, lines 437-439. 'late time point when a majority if viral RNA is distributed throughout the cytoplasm'; what is the evidence for this?
13. The authors have identified several high-NP binding regions (page 6, lines 98-100 and Sp. Tab. 1). However, the significance of these is not discussed. What is the significance of such regions or how NP could be 'enriched' in particular regions?
14. A key paper in the field, Gavazzi et al PNAS 2013, providing experimental evidence for

the importance of a specific interaction between two segments for genome packaging is not referenced.

15. Page 27, line 588. Please add accession number.

Reviewer #3 (Remarks to the Author):

The manuscript entitled "Nucleotide resolution mapping of influenza A virus nucleoprotein-RNA interactions reveals the landscape of viral RNA features required for replication" describes an approach to map regions of viral RNA that are somehow important for viral replication. A main conjecture seems to be that regions of the viral genome that assume stable secondary structures do not bind the nucleoprotein and are important for replication.

As far as I can tell, NP-PAR-CLIP is used to indirectly determine structured regions of the viral genome and I wonder why the authors have not used more direct methods for RNA structure probing. As a way of generating hypotheses about relevant regions to study the CLIP experiment is fine. However, the analysis and validation of the CLIP experiment per se could have been done more extensively. For example, is it clear that the CLIP peaks and troughs are reproducible? I do not see the point of showing the distribution of fragment lengths as this depends on factors such as the genome composition and the sequence specificity of the NP, RNA digestion etc. Similarly, the distance between A-to-G transitions depends on factors that have nothing to do with the binding of the NP. Furthermore, I did not find where the authors defined what they mean by "high confidence binding events". Finally, since the relation between RNA secondary structure and NP binding seems to be a central finding of the manuscript, I would have expected a deeper analysis of it. For example, is there a (anti)correlation between the probability of nucleotides to be paired and their coverage by PAR-CLIP reads? Are there specific structures that prevent binding. etc. ?

Rebuttal to Nature communications:

Reviewer #1 (Remarks to the Author):

Several cryo-EM structures of the influenza A virus (IAV) viral ribonucleoproteins (vRNPs) showed that nucleoprotein (NP) forms a helical scaffold but revealed nothing about the structure of the viral RNA segments (vRNAs) as they were not visible in the density maps. The vRNA structure likely has a profound impact on packaging and genetic reassortment of IAVs and studies addressing the vRNA structure within intact viral particles or purified vRNPs are welcome. Graham et al. used PAR-CLIP to study the genome-wide association of NP with vRNAs in infected cells. Their work represents a very important contribution to the field. There are however, a few points that need to be addressed before publication.

Major points

1) The authors conclude that RNA structure in low-NP regions is required for vRNA packaging (e.g. lines 453-454). While the authors clearly show that mutations in the low NP-regions have an effect on viral replication, the implication of vRNA structure on packaging is suggested, but not convincingly demonstrated. The authors should add a few additional mutants to strengthen this point and modulate their conclusions.

Authors: We understand the reviewer concerns and in response we generated two additional viruses with synonymous mutations that maintained the predicted RNA structure. These viruses, called PB1_{497-561:B} and NS1_{23-68:B} (Figure 4) were not attenuated and had no identifiable defects in genome packaging, thus supporting our conclusion that RNA structures are important for vRNA packaging. Furthermore, we performed RNA thermostability assays on oligomers corresponding to the region on segment 2 (PB1₄₉₇₋₅₆₁). We show that the synonymous mutations, predicted to destabilize the RNA structure, resulted in reduced the thermostability, whereas an oligomer with mutations predicted to preserve wild type structure displayed wild type-like thermal stability. Combined, these data support our overall conclusions. We made text changes to reflect these supporting data.

1.1. Indeed, there is not experimental structural analysis of vRNAs, only secondary structure predictions that were performed on the regions of interest (ROI) (lines 283-287). It is well known that RNA structure predictions are very sensitive to the length of the sequence included in the analysis. Adding or removing a few nucleotides at either end of the ROI would probably significantly affect the analysis. Also, these ROI might form long-range interactions with other parts of the vRNA segment that are completely ignored.

Authors: To provide experimental evidence that RNA structures can occur, we performed RNA thermostability assays on the predicted RNA hairpin in segment 2 (PB1₄₉₇₋₅₆₁). The destabilizing mutations (mutant A) reduced the thermostability (Supplementary Fig 4d) compared to the wild type sequence, whereas the control mutations (mutant B) had no impact on the stability of the RNA. Because mutant A, and not mutant B, is attenuated compared to the wild type virus, it suggests that the RNA structure in this region is important for the virus.

We agree with the reviewer that the structure predictions are sensitive to the length of the RNA sequence. In fact, we noticed that the low-NP binding regions that are not predicted to form a stable structure are often shorter compared to those that do. Extending the low-NP binding regions by a few nucleotides on either end can result in the formation of a stable RNA structure. Since our regions are defined by pre-defined criteria, it is feasible that some of the shorter low-NP binding regions are actually

larger and able to form RNA secondary structures. While we fully appreciate the reviewer reservations that the predictions have limitations, in our study, given that we use multiple length oligomers, including full length segments and viral RNA, the consistency of the collective data strongly support our overall conclusions. Although studies of longer RNAs are important, we believe that these are outside the scope of the current manuscript and will be pursued in the future.

1.2. Only in one case are mutations predicted to affect RNA structure compared to mutations in the same region predicted to have no effect on the structure (mutants NP(22-68:A) and NP(22-68:B)). For all the other low-NP regions analyzed, one cannot exclude that mutations that would not affect the vRNA structure would have the same effect as the mutations that affect vRNA structure. Thus, the authors should analyze the effect of mutations not predicted to affect structure of (two or three) other low-NP binding regions and show that they do not affect replication and packaging.

Authors: We generated two additional mutant viruses containing mutations predicted to not affect the structure of the low-NP binding region (PB1_{497-561:B} and NS1_{23-68:B}). Similar to the NP_{22-68:B} virus, the PB1_{497-561:B} and NS1_{23-68:B} viruses were not attenuated and had no packaging defect. This new information is included in Figure 4 and the result section (line 184) should address this concern.

1.3. The three viruses with mutations in the PB2(1823-1944) region demonstrate that mutations within low-NP regions can affect viral replication without having any negative effect on packaging (lines 184-188, Fig 4f and Fig 6b,c) in contradiction with their conclusion (lines 28-29, 188-191, 453-454). I appreciate the honesty of the authors who included this data in their manuscript even though it doesn't completely fit their main message, but they shouldn't forget it and modulate their conclusion.

Authors: The mutations in PB2₁₈₂₃₋₁₉₄₄ viruses do affect genome packaging (Figure 4), but unlike other low-NP binding regions, the defect is not specific to one or two gene-segments. We applied two different assays to measure genome packaging. The first method quantifies the amount of infectious virus (TCID₅₀) per HA-unit. Defects in genome packaging will result in a decrease in TCID₅₀ per HA-unit. In Figure 4 and 5, we show that the mutations in the low-NP binding region PB2₁₈₂₃₋₁₉₄₄ reduced the TCID₅₀ per HA-unit of the A and B mutant viruses (and to a lesser extent in the C mutant virus). Therefore, we conclude that these mutations affect genome packaging. The second method uses quantitative RT-PCR to identify the relative abundance of each of the segments within a population of purified particles. In the case of PB2₁₈₂₃₋₁₉₄₄, we did not identify one or more gene-segments that were lower compared to the other segments. One possible explanation is that this segment is the central segment in the "7+1" conformation. Changes in the RNA structure of this segment will reduce the packaging of all other gene-segments accordingly producing an equal relative abundance across all segments. Reviewer 2 had a similar comment (#9), therefore we clarified our language in the result section to indicate that the changes in TCID₅₀-to-HA ratio indicate defects in genome packaging. We also modified Table 1 to highlight the effects of these mutations on the virus.

2) The authors conclude that mutations in NP-bound regions have no effect (e.g. lines 29-30). This conclusion is based on four mutants that are all in the NP segment (Fig 2). Mutations of NP-bound regions in other segment(s) would greatly reinforce this conclusion. This is an important point, as this would also reinforce the idea that NP-free and NP-bound regions have different functions.

Authors: We agree with the reviewer's point that analysis of NP bound regions in other gene-segments would reinforce this conclusion. We have added 4 more mutant viruses (two in segment 1 (PB2) and two in segment 2 (PB1)) containing synonymous mutations in NP-bound regions in these gene-segment.

Similar to the mutations in segment 5 (NP), the mutations in NP-bound regions in segment 1 and 2 had no effect virus replication supporting our earlier conclusion that NP-bound regions have no effect. This new information is presented in Figure 4 and in the result section (line 191-192)

3) In order to conclude that 4-SU has no effect on viral replication, the authors should not just show viral titers at 24 hpi (Fig 1b) but replication curves.

Authors: We have performed virus growth curves in 293T cells in the presence and absence of 4-SU. The addition of 4-SU had no effect on the viral titer in the culture supernatant confirming our earlier observation that 4-SU has no effect on viral replication under these conditions. This new information is added in Figure 1b and result section (line 70-71).

4) Analysis of A-to-G mutations:

4.1. These should be A-to-G mutations in the (+) strand, right? The X-linked 4-SU in the (-) (genomic) strand is converted as a G in the (+) strand during reverse transcription. Lines 291-293 are confusing (false?).

Authors: The reviewer is correct and 4-SU induced change is an A-to-G transition in the (+) strand. This is incorrectly stated in the text and figure 1f. We have clarified this in the materials and methods section (line 319) and changed figure 1f.

4.2. Isn't it possible to draw more information from this data than the spacing between the X-linking sites? For instance, it would be interesting to know whether this spacing is the same in the low- and high-NP regions. A map, profile (similar to Supplementary Fig 3) or table of the X-linked site could also be very useful.

Authors: The reviewer raises an interesting point. However, this analysis is complicated by several factors. Foremost, the combined length of the low-NP binding regions is ~10% of the total length of the genome. This leaves us with relatively few data points and thus drawing conclusions difficult. In addition, the average read-depth of the low-NP binding regions is lower compared to the rest of the genome, further complicating the analysis. Thus, we argue that this analysis is currently beyond the scope of the data and manuscript.

Minor points:

1) Introduction, lines 45-45. The paper by Klumpp et al. (1997) EMBO J15, 1248 should be cited.

Authors: We apologize for the omission. This reference is now included in the revised manuscript (line 47).

2) Legend to Supplementary Fig 2 should be more detailed

Authors: We have added more details to the figure legend of Supplementary Figure 2.

3) Supplementary Fig 3 should be much larger, so that readers can retrieve precise information from it

Authors: We have increased the size of panels in Supplementary Figure 3.

4) line 188: Supplementary Fig 5 b-c should read Supplementary Fig 6 b-c

Authors: We apologize for this mistake. It has been corrected in the revised manuscript.

5) The predicted structural effects of the mutations introduced in H7N3 and H1N1 (Fig 5c-d) should be shown in a supplementary figure.

Authors: We agree and have added the predicted structural effects of the mutations in NP₁₄₁₀₋₁₄₉₅ and PB2₁₈₂₃₋₁₉₄₄ in H7N3 and H1N1 to Supplemental Figure 4.

6) Line 241: the wavelength and the apparatus used for X-linking should be indicated.

Authors: These details have been included in Materials and Methods section of the revised manuscript (line 264).

7) Line 254: More details about the preparation of the Illumina sequencing libraries must be provided (amount of RNA, ligation conditions, ...)

Authors: Additional details for the generation of the sequence libraries have been added to the revised manuscript (line 257 - 294).

Reviewer #2 (Remarks to the Author):

In this manuscript by Williams and colleagues use PAR-CLIP to assess the interaction of the influenza virus nucleoprotein (NP) with viral genomic RNA (vRNA) in human cells infected with influenza A/PR/8/34 (H1N1) virus. They find that NP binds to vRNA non-uniformly and without sequence specificity. They identify regions of low NP occupancy as well as regions enriched in NP. Mutation of low NP binding regions resulted in virus attenuation and in most cases this attenuation correlated with a defect in genome packaging. On the other hand, mutations in an NP-bound region had no effect. They extend these studies to a pH1N1 and an avian H7N3 virus by showing that mutations in regions corresponding to the low NP binding regions in PR8 attenuate the replication of these viruses.

Overall, this is the first study that addresses the genome-wide association of NP in an influenza virus. The major finding is that NP is non-uniformly distributed along the vRNAs. Although, this has been suggested by previous studies, this paper provides convincing experimental evidence for this model. Overall, the results are clearly presented and for most parts support the conclusions. The manuscript would benefit from addressing the following points.

Authors: We thank the reviewer for supportive comments and for providing a perspective on the overall value of our study to the field.

1. Page 3, lines 35-38. It is an over-interpretation that refs 1 and 2 solve the structure of the protein components within vRNP. These studies provide little information on the overall structure of the polymerase and propose two contradictory models for the arrangement of NP.

Authors: We have changed the wording on page 3 to more accurately reflect the results and conclusions from these two studies.

2. Fig. 1b. The NP signal is hardly visible in the IF panel and this figure adds little to the manuscript. Either improve or delete.

Authors: We have repeated the experiment and improved the quality of the images in Fig 1b.

3. Page 6, lines 95-100 and Sup. Tab. 1. The labelling in this table and the interpretation of the data is confusing. Bold does not seem to be used consistently and the numbers of low and high NP binding regions in the text and table do not match. Should 'Low (nt)' read 'High (nt)' under Segment 4 on the right? Lines 98-99. 'four high-NP regions that bound RNA were overrepresented ...' is unclear.

Authors: Supplemental Table 1 contained several errors and we apologize for this oversight. In the revised manuscript, we have corrected these mistakes and made additional changes to the supplemental Table 1. Each segment of the viral genome has a 'low' and 'high' NP-binding section. If the segment does not contain a high-NP binding region, it will not be listed. Regions that meet all three criteria are indicated with a *. The total number of regions (all three criteria or 2/3 criteria) have been matched to the numbers depicted in the result section (Line 98).

4. Page 6, lines 102-104. 'Analysis of low-NP binding regions revealed that RNA secondary structures might form in the absence of NP binding in some of these regions.' This statement is not supported by data or references.

Authors: We acknowledge that we did not have direct support for this statement. However, several of our low-NP binding regions overlap with previously identified and studied structures (line 107). Furthermore, RNA structure predictions identified stable structures in a large number of low-NP binding regions. Combined these two data points prompted us to make this statement. To clarify this, we have added the references to this sentence and added the minimal free energy for each of the low- and high-NP binding regions in Supplemental Table 1.

5. Page 7, lines 112-116. NP region 586-608 was included in the analysis as representative of highly bound vRNA. However, this region is not identified as high NP region in Sup. Tab. 1.

Authors: Upon careful review of the data, we acknowledge that this region is not a high-NP binding region according to our strict definition of low- or high NP binding regions, which is $P < 0.01$, > 3 -fold and > 18 nt long. This particular region (NP₅₈₄₋₆₀₄) in segment 5 is $P < 0.01$, > 2 -fold, and 17nt long. Thus, while this region is a potential high-NP binding region, we cannot call it as such based on the current available data. We have removed the mention of the high-NP binding region in the result section and refer to this region as a NP-bound region in the revised manuscript (line 119).

6. Page 7, lines 116-119 and Sup. Fig. 4. There are no experimental data presented to support that the proposed secondary structures form and the introduced mutations disrupt these structures.

Authors: We agree with the reviewer that we had no evidence to support the formation of the predicted secondary structures. To address this comments, we added several new data points to the revised manuscript. The most convincing evidence to support the role of the RNA structure is the addition of more control mutant viruses (PB1_{497-561:B} and NS_{23-86:B}). Viruses containing synonymous mutations in the low-NP binding regions of PB1 and NS, which maintained a RNA structure, were not attenuated and had no defect in genome packaging. These data provide additional evidence that it is the structure and not

the sequence in the low-NP binding regions that is important for the virus. To further support that the attenuating mutations, but not control mutations, altered the secondary structure of the RNA, we performed thermostability assays on RNA oligomers corresponding to the PB1₄₉₇₋₅₆₁ region. The thermal profile of the wild type and one of the mutant RNAs were similar, whereas the RNA containing the structure destabilizing mutations had a different and less stable temperature profile. Combined these data support our findings and conclusions that the RNA structure in low-NP binding regions are important for the virus.

7. Page 8, lines 140-143. NP cannot transcribe and replicate.

Authors: We have corrected this sentence (line 145).

8. Page 9, line 188. 'Supplementary Fig 5 b-c' should read 'Supplementary Fig 6 b-c'.

Authors: We apologize for this mistake. It has been corrected in the revised manuscript.

9. Fig. 4. It is intriguing that the PB2 segment mutations result in the largest attenuation of the virus but there is no detectable defect in vRNA packaging. There is no attempt to determine the mechanism of attenuation for these mutants. Given that out of the three regions analyzed only two are important for coordinating packaging the authors should soften their conclusion (lines 188-191).

Authors: The synonymous structural mutations in the PB2 region do affect genome packaging. We applied two methodologies to assess genome packaging. The first method quantifies the amount of infectious virus (TCID₅₀) per HA-unit. Defects in genome packaging will result in a decrease in TCID₅₀ per HA-unit. As shown in Figures 4 and 5, the mutations in the low-NP binding regions significantly reduce the TCID₅₀ per HA-unit, and thus affect genome packaging. For some of the mutant viruses, we attempted to identify the gene-segment that is missing from the viral particles. For the low-NP binding regions NP₁₄₁₀₋₁₄₉₅ and NP₂₂₋₆₈, we identified specific segments that were significantly reduced in purified viral particles. For the PB2 mutant viruses, we did not identify a specific gene-segment. This was somewhat surprising, but we think that this indicates that segment 1 (PB2) may act as the central gene-segment in the "7+1" vRNP conformation suggested by others. Changes to this segment will alter the coordinated and stoichiometric packaging of all other gene-segments accordingly. Since our analysis is relative to one of the segment (segment 7), all segments will have a similar relative abundance. We have changed the wording in the text and table 1 to clarify that the differences in TCID₅₀-to-HA ratio are indicative of a packaging defect and thus that the mutations in PB2 affect packaging.

10. Fig. 5. Although the introduced mutations result virus attenuation, there is no attempt to link this to a defect in genome packaging. It would be useful to see data on this as well as PAR-CLIP data and an alignment of the relevant RNA regions between PR8, pH1N1 and H7N3 to be able to assess conservation. Secondary structure predictions should also be shown if different from those shown for PR8 in Sup. Fig. 4.

Authors: To link the effects of the synonymous structural mutations to genome packaging, we measured the infectious unit (TCID₅₀)-to-particle ratio for the wild type and mutant H1N1 and H7N3 viruses. A reduction in the amount of infectious virus per HA-unit is indicative of a defect in genome packaging. Similar to the mutations in IAV-PR8 virus, the mutations in NP₁₄₁₀₋₁₄₉₅ and PB2₁₈₂₃₋₁₉₄₄ reduced the TCID₅₀ (infectious unit) per HA-unit. This new information is added in Figure 5 and in the result section (line 228). We have also included the RNA structure predictions, and the effects of the mutations, for the

regions in H7N3 and H1N1 virus in Supplemental Figure 4. CLIP-seq on H7N3 and H1N1, and other strains of influenza virus is a logical and important next step in our analysis. This will determine how conserved low-NP binding regions are between viruses and perhaps identify strain or species specific regions. However, given the magnitude of the work involved and the potential questions that will be addressed in that dataset will be distinct. For these reasons, we believe that such work is outside of the scope of this paper.

11. Page 33 and line 650-656. Legends for panels e and f are switched around.

Authors: corrected as requested.

12. Page 20, lines 437-439. 'late time point when a majority if viral RNA is distributed throughout the cytoplasm'; what is the evidence for this?

Authors: This assumption is based on several studies showing that at 6-8 hours post infection, many of the gene-segments are found in the cytoplasm of the cells. This has been shown using fluorescent RNA probes as well as a PCR assays capable of detecting single RNA molecules inside cells. To support this statement, we have added two references to this sentence that show this.

13. The authors have identified several high-NP binding regions (page 6, lines 98-100 and Sp. Tab. 1). However, the significance of these is not discussed. What is the significance of such regions or how NP could be 'enriched' in particular regions?

Authors: The importance of high-NP binding regions is currently not known. One of the regions in segment 5 (NP₅₈₄₋₆₀₂) is the closest we have come to analyzing these regions, and mutations in this region had no impact on the virus. To address this point, we have included a sentence in the discussion section talking about the high-NP binding regions, and their significance for the virus (Line 490).

14. A key paper in the field, Gavazzi et al PNAS 2013, providing experimental evidence for the importance of a specific interaction between two segments for genome packaging is not referenced.

Authors: We agree with the reviewer and apologize for the omission. This reference is now included in the revised manuscript.

15. Page 27, line 588. Please add accession number.

Authors: Accession numbers will be added once the manuscript is accepted for publication.

Reviewer #3 (Remarks to the Author):

The manuscript entitled "Nucleotide resolution mapping of influenza A virus nucleoprotein-RNA interactions reveals the landscape of viral RNA features required for replication" describes an approach to map regions of viral RNA that are somehow important for viral replication. A main conjecture seems to be that regions of the viral genome that assume stable secondary structures do not bind the nucleoprotein and are important for replication.

As far as I can tell, NP-PAR-CLIP is used to indirectly determine structured regions of the viral genome and I wonder why the authors have not used more direct methods for RNA structure probing.

Authors: We understand the reviewer concerns in that if we set out to identify RNA structures, more direct methods such as SHAPE-seq or DMS-seq may provide direct structural support. However, the goal of the study is to characterize the interaction between NP and the viral RNA. During our investigation, we discovered that the regions with low NP binding regions have a higher propensity to form more stable RNA secondary structures. A subset of these newly identified elements were previously characterized by a variety of methods and shown to be important for the virus (such as the RNA pseudoknot in the NP segment and the RNA structure in the M segment). Therefore, our conclusions from this study that NP binding and RNA secondary structures are important in packaging is strongly supported by the data in our revised manuscript.

That said, the point raised by the reviewer is important and will be the focus of future work. However, these are outside the scope of this manuscript.

To address the point about RNA structures, we have added additional mutant viruses that maintain the RNA structure in low-NP binding regions (PB1_{497-561:B} and NS_{23-86:B}). These viruses are not attenuated further supporting our conclusion that the RNA structure in low-NP binding regions are important for the virus. Finally, we measured the RNA thermostability of the PB1₄₉₇₋₅₆₁ region containing the destabilizing and control mutations. As predicted, only the destabilizing (and attenuating) mutations altered the thermal profile of the RNA, thus supporting our conclusion that RNA structures are important for the virus.

As a way of generating hypotheses about relevant regions to study the CLIP experiment is fine. However, the analysis and validation of the CLIP experiment per se could have been done more extensively. For example, is it clear that the CLIP peaks and troughs are reproducible?

Authors: The data for the CLIP-seq (PAR-CLIP) analysis is derived from four independently repeated assays and we applied statistical analyses to identify high and low-NP binding regions. As such, regions that are consistently low or high will be significant. This is described in the materials and methods section (lines 295-327).

I do not see the point of showing the distribution of fragment lengths as this depends on factors such as the genome composition and the sequence specificity of the NP, RNA digestion etc.

Authors: We agree with the reviewer that the distribution of fragment lengths depends on many factors, most importantly the RNase enzyme and duration of the digestion. However, we feel this is important information as the size of the bound RNA impacts subsequent analysis in the paper, i.e. we set the minimal length of a high and low-NP bound region to 1.5x the average read length (18nt). We did not observe any sequence specificity of NP binding (Figure 1g), therefore we do not think that the length depends on NP sequence specificity or genome composition.

Similarly, the distance between A-to-G transitions depends on factors that have nothing to do with the binding of the NP.

Authors: Again, we agree that there are several factors that determine the distance between U-to-C (or A-to-G in the (+) strand) transitions. The most important factor is the number of uracil residues in a particular region contacted by the viral NP. Because we sample a population, one NP maybe be covalently bound by one uracil, while an adjacent uracil residue will bind another NP molecule. The

distance between these two residues will be calculated as 1-2 nucleotides. We have excluded the 1-base pair distances from the analysis, but this still results in an underestimation of the periodicity between two NP molecules. A second factor that influences the distribution is the low- and high-NP binding regions, where the U-C transition frequency is altered due to NP-binding or lack thereof. Despite these caveats, we believe that this analysis adds to the overall body of work and contributes to our understanding of NP-RNA binding and biology of influenza virus.

Furthermore, I did not find where the authors defined what they mean by "high confidence binding events".

Authors: High confidence binding events are defined by a P-value of <0.01 (calculated from four independently repeated experiments), greater than 3-fold difference between NP-bound and input RNA, and ≥ 18 nucleotides long. Low confidence regions are those that have two out of these three criteria.

Finally, since the relation between RNA secondary structure and NP binding seems to be a central finding of the manuscript, I would have expected a deeper analysis of it. For example, is there a (anti)correlation between the probability of nucleotides to be paired and their coverage by PAR-CLIP reads? Are there specific structures that prevent binding, etc.?

Authors: This is a very interesting point and one we looked at in detail. The short answer is that there is no anti-correlation between the probability of nucleotides to be paired and the coverage by PAR-CLIP. The rationale for this is likely many-fold. First, the probability of nucleotide pairing depends on the length of the RNA sequence that is being considered. Some NP-bound regions, such as NP₁₄₅₋₁₇₅, are predicted to form a RNA structure by *in silico* analysis, but appears to be bound by NP. The reason for this is unknown, but perhaps the location of the RNA structure and its relationship to the NP periodicity is important for the formation of a secondary structure. Additional studies looking at how RNA structures are formed in desperate regions of the viral genome are required to be able to understand how these low-NP binding regions are formed. Also in the interest of prioritizing the work, we have looked at the most significant regions (based on PAR-CLIP). Additional regions may be important, but these are again outside the scope of this study.

REVIEWERS' COMMENTS:

Reviewer #1 (Remarks to the Author):

Overall, the authors made considerable efforts to improve their manuscript. They produced and analyzed several additional mutant viruses that strengthen their conclusions. The authors very satisfactorily answered to all my comments except my major point 1.1. Minor modifications in the text could solve this point (see below) .

The additional data they added to answer this point were performed in vitro on "naked" RNA fragments. Thus, even though this data support the predicted structures, it gives no information about the actual vRNA structure within vRNP. I do realize that obtaining structural information about the vRNA structure within vRNPs would represent a complete study by itself and is far beyond the scope of this study [although such data were recently presented at the ESWI meeting in Riga by the Fodor group and might become available soon]; but in the absence of such data the authors should remain careful in their wording in the results and discussion sections: e.g. I would prefer at line 128 "those destabilizing a predicted stem-loop structure", line 132 "... synonymous mutations designed to maintain the predicted secondary structure", line 163 "... mutations designed to maintain the predicted 3' stem-loop structure" similar changes should be introduced at lines 183, 185, 226, 227, 485 etc.

Minor point: references 12 and 14 correspond to the same paper.

The present work represents an important step forward compared to the recent CLIP-seq paper by the Lakdawala group and in my opinion it fully deserves publication in Nature Communications.

Reviewer #2 (Remarks to the Author):

In my opinion the authors have satisfactorily addressed the reviewers' concerns and the manuscript has been improved. I have no further comments (apart from a minor point: correct text at lines 489-493).

Reviewer #3 (Remarks to the Author):

The authors have partially addressed my comments. I still do not see the relevance of the fragment length distribution or that of the distance between A-to-G mutations. The authors state that the protein has no sequence specificity and rather pursue the argument that the RNA secondary structure determines binding. But then how does the distance between A-to-G mutations reveal anything about the number of NP monomers that are bound to an RNA fragment? Furthermore, in what sense is there any periodicity of binding as the authors mention in their response?

Rebuttal to Reviewers.

Reviewer #1 (Remarks to the Author):

Overall, the authors made considerable efforts to improve their manuscript. They produced and analyzed several additional mutant viruses that strengthen their conclusions. The authors very satisfactorily answered to all my comments except my major point 1.1. Minor modifications in the text could solve this point (see below).

Authors: We thank the reviewer for the positive assessment of our manuscript.

The additional data they added to answer this point were performed in vitro on “naked” RNA fragments. Thus, even though this data support the predicted structures, it gives no information about the actual vRNA structure within vRNP. I do realize that obtaining structural information about the vRNA structure within vRNPs would represent a complete study by itself and is far beyond the scope of this study [although such data were recently presented at the ESWI meeting in Riga by the Fodor group and might become available soon]; but in the absence of such data the authors should remain careful in their wording in the results and discussion sections: e.g. I would prefer at line 128 “those destabilizing a predicted stem-loop structure”, line 132 “... synonymous mutations designed to maintain the predicted secondary structure”, line 163 “ ... mutations designed to maintain the predicted 3’ stem-loop structure” similar changes should be introduced at lines 183, 185, 226, 227, 485 etc.

Authors: We have changed our wording throughout the manuscript to make clear that the RNA structures are predicted structures.

Minor point: references 12 and 14 correspond to the same paper.

Authors: References 12 and 14 have the same first author, but are not the same papers.

The present work represents an important step forward compared to the recent CLIP-seq paper by the Lakdawala group and in my opinion it fully deserves publication in Nature Communications.

Authors: We thank the reviewer for the positive assessment of our manuscript.

Reviewer #2 (Remarks to the Author):

In my opinion the authors have satisfactorily addressed the reviewers’ concerns and the manuscript has been improved. I have no further comments (apart from a minor point: correct text at lines 489-493).

Authors: We have corrected the text at the indicated lines.

Reviewer #3 (Remarks to the Author):

The authors have partially addressed my comments. I still do not see the relevance of the fragment length distribution or that of the distance between A-to-G mutations. The authors state that the protein has no sequence specificity and rather pursue the argument that the RNA secondary structure determines binding. But then how does the distance between A-to-G mutations reveal anything about

the number of NP monomers that are bound to an RNA fragment? Furthermore, in what sense is there any periodicity of binding as the authors mention in their response?

Authors: We agree with the reviewer that the distribution of fragment lengths depends on many factors, most importantly the RNase enzyme and duration of the digestion. However, we feel this is important information as the size of the bound RNA impacts subsequent analysis in the paper, i.e. we set the minimal length of a high and low-NP bound region to 1.5x the average read length (18nt). We did not observe any sequence specificity of NP binding (Figure 1g), therefore we do not think that the length depends on NP sequence specificity or genome composition.

The distance between U-to-C (or A-to-G in the (+) strand) transitions also depends on several factors including, as stated by the reviewer, RNA secondary structures. It also depends on the number of uracil residues in a given NP footprint. Because of these caveats and the reservation by this reviewer, we have decided to remove the figure (and associated text) from the manuscript.